# Molecular basis for N-terminal acetylation by human NatE and its modulation by HYPK

Sunbin Deng [1,2], Nina McTiernan [3], Xuepeng Wei [2,4], Thomas Arnesen[3,5,6] & Ronen Marmorstein [1,2,4]*

The human N-terminal acetyltransferase E (NatE) contains NAA10 and NAA50 catalytic, and NAA15 auxiliary subunits and associates with HYPK, a protein with intrinsic NAA10 inhibitory activity. NatE co-translationally acetylates the N-terminus of half the proteome to mediate diverse biological processes, including protein half-life, localization, and interaction. The molecular basis for how NatE and HYPK cooperate is unknown. Here, we report the cryo-EM structures of human NatE and NatE/HYPK complexes and associated biochemistry. We reveal that NAA50 and HYPK exhibit negative cooperative binding to NAA15 in vitro and in human cells by inducing NAA15 shifts in opposing directions. NAA50 and HYPK each contribute to NAA10 activity inhibition through structural alteration of the NAA10 substrate-binding site. NAA50 activity is increased through NAA15 tethering, but is inhibited by HYPK through structural alteration of the NatE substrate-binding site. These studies reveal the molecular basis for coordinated N-terminal acetylation by NatE and HYPK.

[1] Department of Chemistry, University of Pennsylvania, Philadelphia, PA 19104, USA. [2] Abramson Family Cancer Research Institute, Perelman School of Medicine, University of Pennsylvania, Philadelphia, PA 19104, USA. [3] Department of Biomedicine, University of Bergen, Bergen, Norway. [4] Department of Biochemistry and Biophysics, Perelman School of Medicine, University of Pennsylvania, Philadelphia, PA 19104, USA. [5] Department of Biological Sciences, University of Bergen, Bergen, Norway. [6] Department of Surgery, Haukeland University Hospital, Bergen, Norway. *email: marmor@upenn.edu

Protein N-terminal acetylation is one of the most ubiquitous covalent modifications among eukaryotes, occurring on ~80% of human proteins[1,2]. This modification affects many protein functions, including protein half-life, folding, complex formation, and localization[1,3–10]. Protein N-terminal acetylation is considered an irreversible process and is catalyzed by N-terminal acetyltransferases (NATs). In human, seven different NATs with different subunit composition and substrate preferences have been identified to date, including hNatA through hNatH[11], with the exception that NatG is only present in chloroplasts of plant cells[12]. NatA, NatB, and NatC are three major NATs, each consisting of a catalytic subunit and one (NatA and NatB) or two (NatC) auxiliary subunits[11,13]. The NatE complex contains NatA and an additional catalytic subunit, NAA50[14,15]. NatD, NatF, and NatH function in the absence of an auxiliary subunit. NatA acetylates non-methionine containing substrates harboring small N-terminal residues (alanine, serine, threonine, cysteine, and valine)[2,16]. NatB/C/E/F modify proteins with N-terminal methionine residues: NatB acetylates methionine followed by a negatively charged residue[17,18], while NatC/E/F acetylate methionine followed by hydrophobic/amphipathic residues[5,15,19–22]. NatD and NatH exhibit highly selective substrate recognition patterns; NatD acetylates only the N-terminus of histones H2A and H4[23,24], while NatH acts solely on processed actins[25,26]. NatA through NatE function predominantly co-translationally through association with the ribosome[27], while NatF anchors onto the Golgi membrane to acetylate transmembrane proteins[20,28,29].

About one-half of the human N-terminal acetylome is mediated by hNatA. Mutations in either the hNAA10 catalytic or hNAA15 auxiliary subunits have been correlated with a broad spectrum of pathologies, including intellectual disabilities, developmental delay, autism spectrum disorders, craniofacial dysmorphology, congenital cardiac anomalies, and Ogden syndrome[30–39]. Aberrant NatA activity has also been correlated with neurodegenerative disorders and cancer although a causative role of NatA is less clear[40]. NatA forms complexes with at least two other proteins, NAA50 and the chaperone protein Huntingtin-interacting protein K (HYPK)[14,41–44]. The trimeric hNAA10/hNAA15/hNAA50 complex is referred to as hNatE, and we refer to the tetrameric hNAA10/hNAA15/hNAA50/HYPK complex as hNatE/HYPK. Previous studies have demonstrated that HYPK may be important for cellular NatA activity and that NAA50 and NatA can affect each other's function[16,41,44–46]. Previous studies also demonstrated that hNatA can physically associate with hNAA50 (ref. [14]) and HYPK (ref. [41]), separately. Moreover, the crystal structures of hNatA with HYPK and related pull-down experiments indicated that HYPK and hNAA50 binding to hNatA might be mutually exclusive[16]. However, another study suggested that HYPK and NatE form a tetrameric complex in *Drosophila* cells[45].

To delineate the mechanistic basis for how the NAA10 and NAA50 catalytic subunits of the NatE complex coordinate function and how this is regulated by HYPK, we characterized the human NatE and NatE/HYPK complexes biochemically, structurally, and in cells. We show that hNAA50 and HYPK exhibit negative cooperative binding to hNatA in vitro and in human cells, by inducing hNAA15 shifts in opposing directions. We show that hNAA50 and HYPK both mediate hNAA10 inhibition through structural alteration of the hNAA10 substrate-binding site. We show that hNatE is about tenfold more active than hNAA50, likely due to a reduced entropic cost for substrate-binding through hNatA tethering, but is inhibited by HYPK through structural alteration of the hNatE substrate-binding site. Taken together, these studies reveal the molecular basis for coordinated N-terminal acetylation by the hNAA10 and hNAA50 catalytic subunits of NatE and its modulation by HYPK.

## Results

**hNatE and HYPK form a tetrameric complex.** To determine if human NAA50 and HYPK can simultaneously bind to NatA to form a hNatE/HYPK complex, we prepared the recombinant hNatA/HYPK complex from insect cells and mixed it with recombinant hNAA50 for analysis on size-exclusion chromatography (Fig. 1a—left). Analysis of the peak fractions on sodium dodecyl sulfate–polyacrylamide gel electrophoresis (SDS–PAGE) revealed that the four protein components (hNAA10, hNAA15, hNAA50, and HYPK) co-eluted in a single major peak with excess hNAA50 eluting in later fractions (Fig. 1a). This result demonstrates that HYPK and hNAA50 can bind to hNatA simultaneously to form a tetrameric hNatE/HYPK complex.

**HYPK and hNAA50 display negative cooperative binding to hNatA.** Previous studies demonstrated that both human HYPK and hNAA50 bind tightly to hNatA, with dissociation constants in the nanomolar range[16,46]. We set out to determine if the HYPK- and hNAA50 binding properties to hNatA are altered in the context of the tetrameric complex. Using fluorescence polarization (FP) assays, we observed that hNAA50 bound to hNatA with a $K_d$ of $46 \pm 8.8$ nM (Fig. 1b), consistent with previous results. However, in the presence of maltose-binding protein (MBP) tagged HYPK (MBP-HYPK), this binding $K_d$ decreased about threefold to $127 \pm 13$ nM (Fig. 1b). This data demonstrates that HYPK can negatively affect the stability of the hNatE complex by about threefold. Based on this data, we hypothesized that in the presence of hNAA50, the binding affinity between hNatA and HYPK would also decrease. Using isothermal titration calorimetry (ITC), we determined that the $K_d$ between hNatA and MBP-HYPK was $29.9 \pm 16.7$ nM (Fig. 1c), which is consistent with the previously reported $K_d$ in the thermophilic fungus *Chaetomium thermophilum*[44]. The reason for the apparent two-binding transition when HYPK is titrated into hNatA is unknown. We further determined the $K_d$ between HYPK and hNatE, and observed that the affinity was $154.1 \pm 55.3$ nM, approximately fivefold weaker than HYPK binding to hNatA (Fig. 1d). A control ITC run of MBP titrated into hNatA failed to show binding (Supplementary Fig. 1). Taken together, these data indicate that HYPK and hNAA50 display negative cooperative binding with respect to hNatA.

**hNAA50 and HYPK inhibit hNatA activity, and HYPK is dominant.** While our previous studies demonstrated that yeast and human NatA have slightly elevated catalytic efficiencies in the presence of NAA50, we set out to make a more direct comparison of NatA activity as a function of added NAA50 vs. HYPK. To do this, we used an in vitro acetylation assay using a cognate SASE NatA substrate (Ser-Ala-Ser-Glu; for full sequence see Methods) to compare the hNatA catalytic activity as a function of added hNAA50 or MBP-HYPK. This comparison revealed that while MBP-HYPK showed ~90% inhibition of hNatA activity at an MBP-HYPK concentration beyond ~100 nM, hNAA50 only decreased hNatA activity by ~50% at concentrations between ~100 nM and 1 μM (Fig. 2a), consistent with the previous data[16,46]. Titrating both MBP-HYPK and hNAA50 into hNatA showed a similar degree to hNatA inhibition as did titrating in MBP-HYPK alone, illustrating that HYPK inhibition of hNatA overrides the incomplete inhibitory effect of hNAA50 (Fig. 2a).

**HYPK inhibits hNatE activity.** Given that HYPK was shown to be a negative regulator for hNatA acetylation activity[16,44], we asked if it also inhibited the activity hNatE. To do this, we used the same in vitro acetyltransferase activity assay but employed a cognate

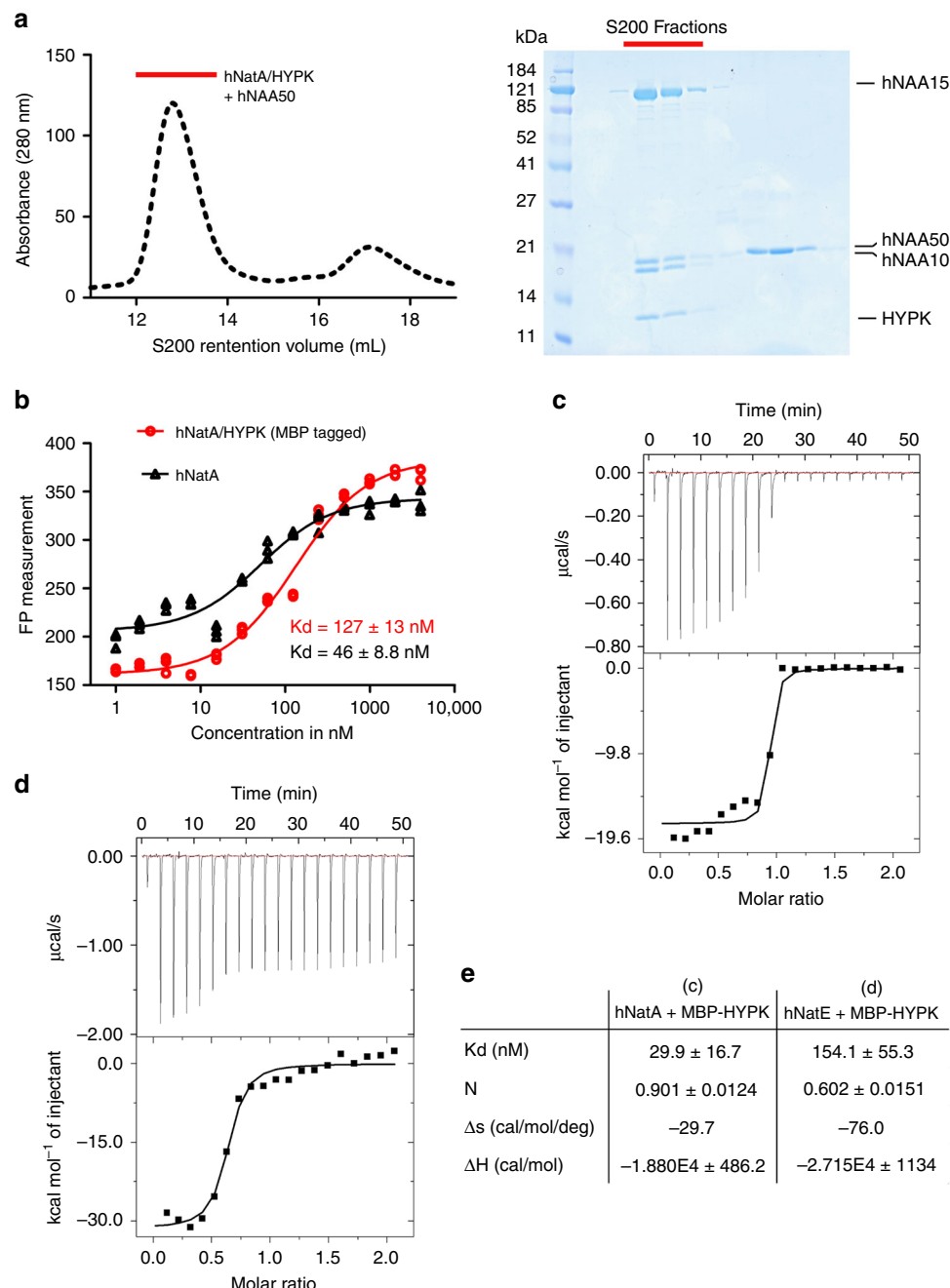

**Fig. 1 HYPK and hNatE form a tetrameric complex. a** Left—gel filtration elution profile of the hNatA/HYPK complex with excess hNAA50, using a Superdex S200 column. Right—Coomassie-stained SDS–PAGE of peak fractions. Red bars indicates the peak complex. **b** Fluorescence polarization assays with either hNatA or hNatA/MBP-HYPK titrated into fluorescein-5-maleimide-labeled hNAA50. The data is fit to calculate a dissociation constant ($K_d$). Replicates were shown in the curve with $n = 3$ independent experiments. Source data are provided as a Source Data file. **c** Representative ITC curve of MBP-HYPK titrated into hNatA. The calculated $K_d$ is indicated. **d** Representative ITC curve of MBP-HYPK titrated into hNatE. **e** The ITC fitting information and calculated $K_d$ is provided for curves **c** and **d**.

MLGP hNatE substrate (Met-Leu-Gly-Pro; for full sequence see Methods) in the absence or presence of HYPK, as well as a hNAA50 only control. This analysis revealed that while hNatE was more active than hNAA50, this improvement was essentially nullified with the addition of HYPK (Fig. 2b–d). A full kinetic analysis of hNAA50, hNatE, and hNatE/HYPK revealed the kinetic parameters responsible for this modulation of hNAA50 function. Specifically, we found that hNatE displayed ~8.6-fold decrease of $K_m$, and ~1.1-fold decrease of $k_{cat}$, with an overall ~7.7-fold increase of catalytic efficiency, compared to hNAA50 (Table 1). These findings

are in general agreement with the previous studies with the orthologous yeast proteins[46]. A comparison of the activities of hNatE and hNatE/HYPK revealed that in the presence HYPK, hNatE displayed an ~1.3-fold decrease in $K_m$, and an ~3.8-fold decrease in $k_{cat}$, with an overall ~2.9-fold decrease in catalytic efficiency (Table 1). This data suggests that HYPK can compromise hNatE activity. Taken together, while hNatA significantly enhances the catalytic efficiency of hNatE, HYPK binding to hNatE largely nullifies this effect, demonstrating that HYPK can indirectly inhibit hNatE activity.

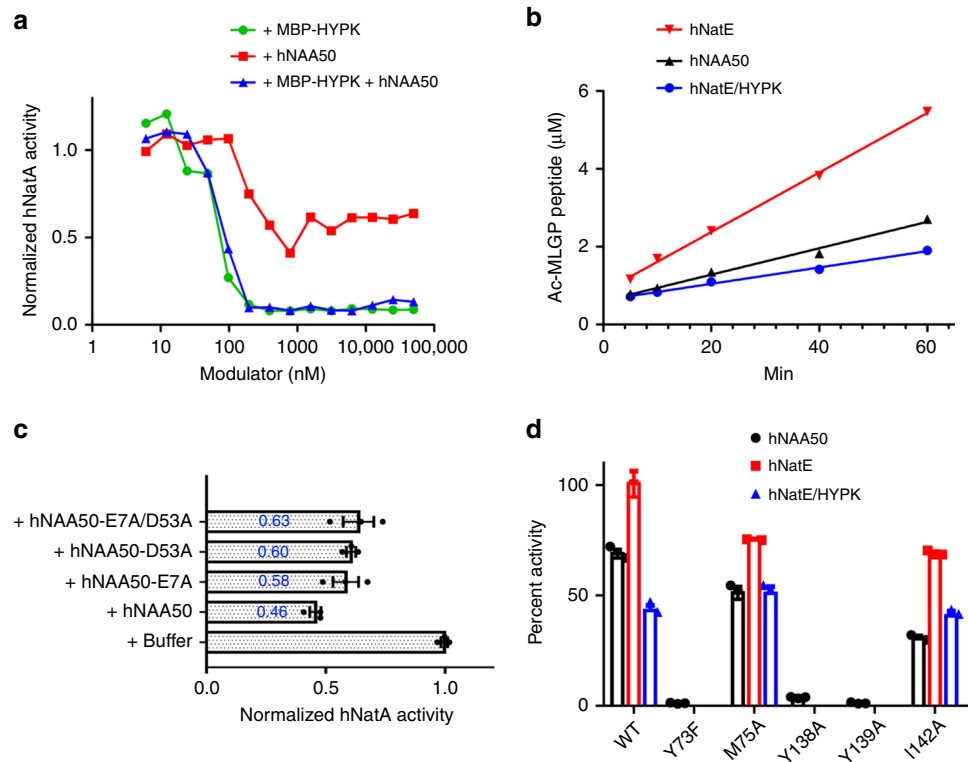

**Fig. 2 HYPK binding negatively affects hNatE acetylation activity. a** Either MBP-HYPK (green), hNAA50 (red), or both (blue) are titrated into hNatA (100 nM) to evaluate their modulatory effect on hNatA activity against an SASE peptide substrate. A best line is drawn through the data points for ease of visualization. Source data are provided as a Source Data file. **b** Comparison of time course acetylation activity of hNAA50 (black), preformed hNatE (red), and preformed hNatE/HYPK (blue; all 500 nM) against an MLGP peptide substrate. Source data are provided as a Source Data file. **c** hNatA acetylation activity against an SASE peptide with addition of buffer, wild type (WT), or hNAA50 mutants. Errors were reported in SEM with $n = 3$ independent experiments. Source data are provided as a Source Data file. **d** Activity of hNAA50 WT and mutants against MLGP peptide was tested, either alone (black), in the context of hNatE (+hNatA; red), or in the context of preformed hNatE/HYPK (+hNatA/HYPK; blue). Data were normalized to WT hNatE activity as 100%, represented as mean ± SD, $n = 3$ independent experiments. Source data are provided as a Source Data file.

**Table 1 Catalytic parameters for hNAA50, hNatE, and hNatE/HYPK.**

|  | $k_{cat}$ (min$^{-1}$) | $K_m$ (μM) | $k_{cat}/K_m$ (μM$^{-1}$ min$^{-1}$) |
|---|---|---|---|
| hNAA50* | 3.45 | 831.5 | $4.15 \times 10^{-3}$ |
| hNatE* | 3.09 | 96.2 | $3.21 \times 10^{-2}$ |
| hNatE/HYPK | 0.81 | 73.1 | $1.11 \times 10^{-2}$ |

*Calculation of kinetic parameters for hNAA50 and hNatE is based on an independent replicate of experiments previously reported[46]. Source data are provided in the Source Data file.

**hNatA forms competing complexes with hNAA50 and HYPK in cells**. To further confirm the existence of the tetrameric hNatE/HYPK complex observed in vitro in human cells, we immuno-precipitated hNAA15 (c-terminal V5 tagged) from HeLa cells to evaluate the associated proteins. Mass spectrometry (MS) analysis of this sample revealed that endogenous hNAA10, hNAA50, and HYPK were all co-immunoprecipitated with NAA15-WT-V5, in agreement with the tetrameric hNatE/HYPK complex formation (Fig. 3a, Supplementary Fig. 2).

In prior studies, we reported that a hNAA15-T406Y mutant can disassociate hNAA50 from hNatA in vitro[46], while a hNAA15-L814P mutant is defective for HYPK inhibition and reduces hNatA thermostability[37]. Based on these observations, we also carried out a MS analysis of the V5 immunoprecipitates of the hNAA15-T406Y-V5 and hNAA15-L814P-V5 mutants to evaluate the effect of the

mutation's on formation of the hNatE/HYPK tetrameric complex (Supplementary Data 1). For relative quantification of the hNatA proteins, the intensity-based absolute quantification (IBAQ) intensities of the hNatA components in each sample were normalized to the IBAQ intensity of hNAA15 in the respective sample and to the corresponding protein in the (wild-type) WT sample (Fig. 3a, Supplementary Data 1). This analysis demonstrated that hNAA15-T406Y-V5 and hNAA15-L814P-V5 had reduced binding of hNAA50 and HYPK respectively, indicating that the mutations negatively affected the ability of hNAA15 to bind hNAA50 and HYPK, respectively. Furthermore, it appeared that the reduced binding of HYPK or hNAA50 resulted in greater binding of the other component: hNAA15-T406Y-V5 bound less hNAA50, but more HYPK than hNAA15-WT, while hNAA15-L814P-V5 bound less HYPK and more hNAA50 than hNAA15-WT (Fig. 3a, Supplementary Data 1, Supplementary Fig. 2). These observations are in agreement with the in vitro FP and ITC assays, demonstrating that HYPK and hNAA50 display negative cooperative binding with respect to hNatE. Taken together, these cellular studies agree with the in vitro studies by confirming the existence of the tetrameric hNatE/HYPK complex, and indicating that HYPK and hNAA50 display negative cooperative binding with respect to hNatE.

The intrinsic hNatA catalytic activities of the hNAA15-T406Y-V5 and hNAA15-L814P-V5 variants were also tested in a Nt-acetylation assay and western blot (Fig. 3b, Supplementary Fig. 3). Importantly, equal levels of hNAA10 was pulled out with the NAA15 mutants from the lysates supporting that none of the

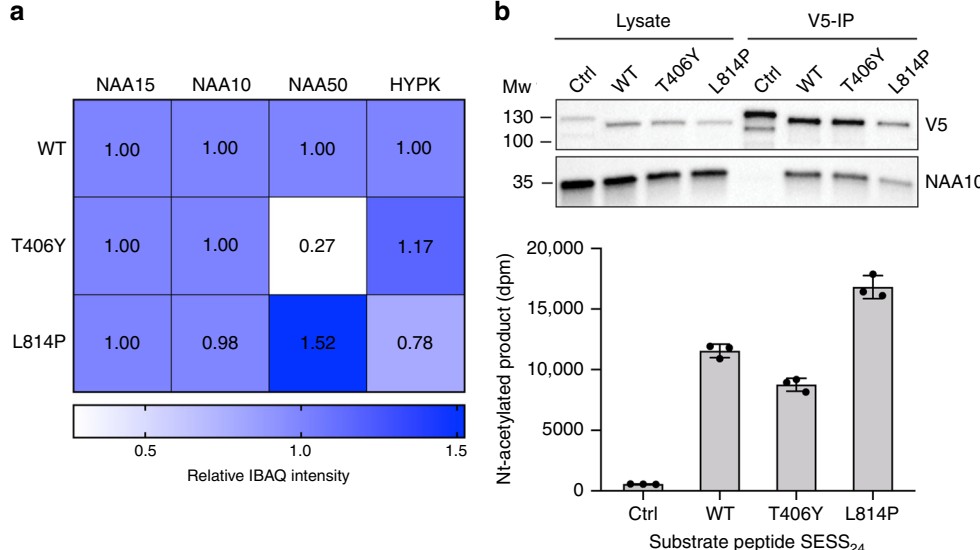

**Fig. 3 hNatA forms competing complexes with hNAA50 and HYPK. a** Heat map showing the mean relative IBAQ intensity of NatA components from three independent MS analyses of immunoprecipitated NAA15 variants, NAA15-WT-V5, NAA15-T406Y-V5, and NAA15-L814P-V5. The IBAQ intensities of each component were normalized to the IBAQ intensity of NAA15 in the respective sample and to the corresponding NatA WT protein. Source data are provided as a Source Data file. **b** Western blot analysis and NatA Nt-acetylation assay of V5-immunoprecipiated NAA15 variants. The measured DPM signal for each reaction was normalized to the corresponding V5-band in the IP. The immunoprecipitation and activity assay were performed in $n = 3$ independent experiments, each with three technical replicates (indicated by dot plots) per assay. Source data are provided as a Source Data file.

mutants affect hNAA10 binding. The hNAA15-T406Y-V5 hNatA complex displayed a decreased catalytic activity toward the hNatA substrate SESS_24 compared to WT hNatA. This could be explained by an increased binding and inhibitory effect of HYPK and/or due to the reduced binding of hNAA50. The hNAA15-L814P-V5 hNatA complex showed an increased catalytic activity compared to WT hNatA, which is consistent with the in vitro findings of reduced binding of HYPK and consequently less inhibition of the catalytic activity and/or an increased binding of hNAA50.

**hNatE structure reveals molecular basis for enzyme crosstalk**. To determine the molecular basis for interaction and enzymatic crosstalk between human NAA10 and NAA50 enzyme subunits within the NatE complex, we determined the single-particle cryogenic electron microscopy (cryo-EM) structure of the hNatE complex in the presence of inositol hexakisphosphate IP6 and bi-substrate analogs of both hNatA and hNAA50. The model of the human complex was rigid body fitted with the coordinates of the crystal structure of hNatA (PDB: 6C9M) and hNAA50 (PDB: 3TFY), with all ligands removed. After refinement of the model, we observed clear density of IP6 and acetyl-CoA cofactors in each catalytic subunit, but the peptide portion of the bi-substrate inhibitors was not resolved. The structure was refined to an overall resolution of 3.0 Å, with most residues of hNAA10 and hNAA50 subunits resolved to 2.5 Å (Supplementary Figs. 4, 5, 6, and 7). The refinement statistics can be found in Table 2. The N-terminal 112 residues of hNAA15 subunit were not modeled due to a lack of traceable cryo-EM density for this region.

Consistent with the previous studies with the orthologous yeast proteins[46], hNAA50 predominantly interacts with hNAA15 (α21, α22, α23, and α24), and makes relatively few interactions with hNAA10 (Fig. 4a), burying a solvent excluded surface of 34,316 Å² with hNAA15, and 9100 Å² with hNAA10. Based on the structure of the ScNatE complex, a hNAA15-T406Y mutant was shown to disrupt the hNatA-NAA50 interaction[46]. Indeed, we find that hNAA15-Thr406 is located at the center of the hNAA50-hNAA15 interface, making hydrogen bond interactions with the backbone carbonyl group of hNAA50-Ala55 (Fig. 4b). Additional hydrogen bonds are formed between the backbone carbonyl of hNAA50-His14 with the backbone amide of hNAA15-Thr439, between hNAA50-Arg21 and the backbone carbonyl of hNAA15-Pro405, and between hNAA50-Asn52 and hNAA15-Thr371 (Fig. 4b). hNAA50-Arg21 also makes direct ionic interaction with hNAA15-Glu433, which is absent in yeast (Fig. 4b). It is possible that this ionic interaction underlies the increased salt sensitivity of the human vs. yeast NatE complex[46]. Interestingly, we do not observe significant hydrophobic interactions at the hNAA15-hNAA50 contact interface.

hNAA50-hNAA10 interactions are less extensive than hNAA50-hNAA15, yet still significant. Interactions are observed between hNAA50-Glu7 and hNAA10-Arg116, and between hNAA50-Asp53 and hNAA10-Arg83 (Fig. 4c). This structural observation highlights the recent identification of a missense mutation of hNAA10 R83H that exhibits decreased acetylation activity in boys with intellectual disability and developmental delay[38]. It is noteworthy that hNAA10-Arg83 also makes interaction with the 3′ phosphate of the acetyl-CoA molecule bound to hNAA10 (Fig. 4c).

To assess the importance of the NatA-NAA50 interactions on the catalytic crosstalk of the two enzymes, we prepared hNAA50-E7A, -D53A and -E7A/D53A mutants and evaluated their effect on hNatA activity (Fig. 2c). We found that the single and double mutants were able to restore between 12–17% of hNatA activity relative to the activity of hNatA bound to WT-hNAA50 (Fig. 2c). These results demonstrate that the observed hydrogen bond interactions between hNatA and hNAA50 make a small, albeit significant, contribution to the catalytic crosstalk between the two enzymes.

An overlay of the *Saccharomyces cerevisiae* (PDB: 6O07) and human NatE structures reveals a high degree of structural conservation with a root-mean square deviation (RMSD) of 1.517 Å (over 622 common Cα atoms). However, it is noteworthy that when bound to NatA, we observe that NAA50 shifts significantly closer to NAA10 in the human over the yeast complex, resulting in more significant and intimate NAA10-NAA50 interactions in the

**Table 2 Cryo-EM data collection, refinement, and validation statistics.**

|  | hNatE/HYPK EMD-20501 PDB: 6PW9 | hNatE EMD-20442 PDB: 6PPL |
|---|---|---|
| **Data collection and processing** |  |  |
| Magnification | 36,000 | 165,000 |
| Voltage (keV) | 200 | 300 |
| Electron exposure (e/Å²) | 40 | 40 |
| Defocus range (μm) | −1.5 to −3.0 | −1.5 to −3.0 |
| Pixel size (Å) | 1.169 | 0.832 |
| Symmetry imposed | C1 | C1 |
| Initial particles(no.) | 477,608 | 1,229,331 |
| Final particles (no.) | 168,536 | 353,541 |
| Map resolution (Å) | 4.03 | 3.02 |
| FSC threshold | 0.143 | 0.143 |
| Map resolution range (Å) | 3.5–5.0 | 2.5-4.5 |
| **Refinement** |  |  |
| Initial model used (PDB code) | 6C95 and 3TFY | 6C9M and 3TFY |
| Model resolution (Å) | 4.3 | 3.1 |
| FSC threshold | 0.5 | 0.5 |
| Model resolution range (Å) | — | — |
| Map sharpening B factor (Å²) | −154.684 | −119.78 |
| **Model composition** |  |  |
| Non-hydrogen atoms | 8836 | 8155 |
| Protein residues | 1075 | 980 |
| Ligands | 2 | 3 |
| **B factors (Å²)** |  |  |
| Protein | 91.05 | 38.81 |
| Ligand | 68.44 | 46.42 |
| **RMSD** |  |  |
| Bonds lengths (Å) | 0.004 | 0.006 |
| Bond angles (°) | 0.990 | 0.811 |
| **Validation** |  |  |
| MolProbity score | 1.52 | 1.96 |
| Clash score | 4.71 | 3.88 |
| Poor rotamers (%) | 1.49 | 0.42 |
| **Ramachandran plot** |  |  |
| Favored (%) | 96.53 | 95.65 |
| Allowed (%) | 100 | 100 |
| Disallowed (%) | 0 | 0 |

human complex (Fig. 5a). α3 and β7 of hNAA50 shift toward hNAA10 ~10 Å and 11 Å, respectively (Fig. 5a). Such significant shift in hNAA50 position relative to hNatA is surprising and may reflect a functional importance when these proteins are associated with the ribosome (Discussion section).

To understand other structural contributions that hNAA50 might have on hNatA activity, we superimposed the hNatE structure with the hNatA crystal structure (PDB: 6C9M). We found that hNAA15 and hNAA10 in the ternary hNatE complex displayed an RMSD of 0.747 Å (over 624 common Cα atoms), and 0.897 Å (over 141 common Cα atoms), respectively (Fig. 5b). The hNAA10 β6–β7 loop displayed about a 4 Å shift toward hNAA50 in the hNatE complex relative to the hNatA complex (Fig. 5b). Given that the β6–β7 loop of hNAA10 plays an important role in peptide substrate recognition, we propose that this shift in position also contributes to the inhibitory effect that hNAA50 binding has on hNatA activity.

To understand the molecular basis for why hNatA binding increases the catalytic efficiency of hNatE by nearly eightfold (Fig. 2b, Table 1), we superimposed the hNAA50 crystal structure

(PDB: 3TFY) with the hNAA50 subunits of the hNatE structure. Remarkably, we did not observe any significant structural changes in hNAA50 with an overall RMSD for all atoms of 0.437 Å (Fig. 5b). Notably, the β6–β7 and α1–α2 peptide substrate-binding loops of hNAA50 within the hNatE complex super-imposed well with hNAA50 bound to CoA and substrate peptide (Fig. 5b). Based on this observation, we hypothesize that the increased activity of hNatE is due to a reduced entropic cost for substrate binding to the hNAA50 subunit due to hNatA tethering. This is consistent with the observations of Wand and colleagues for other protein systems[47].

**hNatE/HYPK structure reveals negative cooperative mechanism.** To obtain a molecular understanding of how hNAA50 and HYPK can both bind hNatA, we determined the structure of the hNatE/HYPK complex by cryo-EM, which we were able to resolve to an overall resolution of 4.0 Å (Supplementary Figs. 4–7). The starting structure was modeled using the hNatA/HYPK (PDB: 6C95) and hNAA50 (PDB: 3TFY), which were placed into the cryo-EM map through rigid body fitting, followed by adjustment and refinement. The refinement statistics can be found in Table 2.

In the tetrameric structure, we observe that both hNAA50 and HYPK simultaneously dock onto the same binding regions on hNatA, as previously identified in the absence of the other protein[16]. While both hNAA50 and HYPK contact both subunits of hNatA, no direct interactions are observed between hNAA50 and HYPK (Fig. 6a). HYPK binds NatA mainly through interactions with the hNAA15 subunit of hNatA: only the α1 helix of HYPK interacts with hNAA10, while the α2 and C-terminal UBA domain (α3, α4, and α5 helices) of HYPK interact with hNAA15 (Fig. 6a). Previous studies demonstrated that the UBA domain and α2 helix play key roles in hNatA binding, while α1 is essential for hNatA activity inhibition[16,44].

Over the HYPK and hNatA interaction interface within the tetrameric complex, we observe polar interactions between HYPK-Glu74 and hNAA15-Tyr158, between the backbone carbonyl of HYPK-Thr100 and hNAA15-Lys687, between the backbone carbonyl of HYPK-Asn129 and hNAA15-Arg697, and between HYPK-Asn129 and hNAA15-Lys696, which were all observed similarly in the hNatA/HYPK crystal structure[16] (Fig. 6b). However, HYPK-Glu103, hNAA15-Lys685, and hNAA15-Glu655 which form a salt bridge in the hNatA/HYPK structure do not show a similar contact in the tetrameric hNatE/HYPK complex[16] (Fig. 6b). We speculate that the absence of this interaction might contribute to the slightly weaker affinity of HYPK for hNatE over hNatA.

hNAA15 and hNAA50 also make similar interactions in the absence or presence of HYPK. The backbone carbonyl of hNAA50-His14 hydrogen bonds with the backbone amide of hNAA15-Thr439, and hNAA50-Asn52 hydrogen bonds with hNAA15-Thr371 (Fig. 6c). Interestingly, however, a key residue that stabilizes the hNatA-hNAA50 interaction, hNAA15-Thr406 shifts from in interaction with the carbonyl of hNAA50-Ala55 to a hydrogen bond to the hNAA50-Gln18 side chain in the tetrameric complex (Fig. 6c). An additional H-bond is formed between hNAA50-Gln18 and the backbone carbonyl of hNAA15-Thr406 (Fig. 6c). hNAA50-Arg21, a residue that makes extensive contacts to NAA15 in the hNatA/NAA50 complex (Fig. 4b) is only contacting hNAA15-Glu433 in the tetrameric complex (Fig. 6c). In addition, while hNAA50-Asp53 and hNAA10-Arg83 maintain interaction as observed in the hNatA/hNAA50 complex, hNAA50-Glu7 and hNAA10-Arg116 lose interaction in the tetrameric complex (Fig. 6d). These observations correlate with

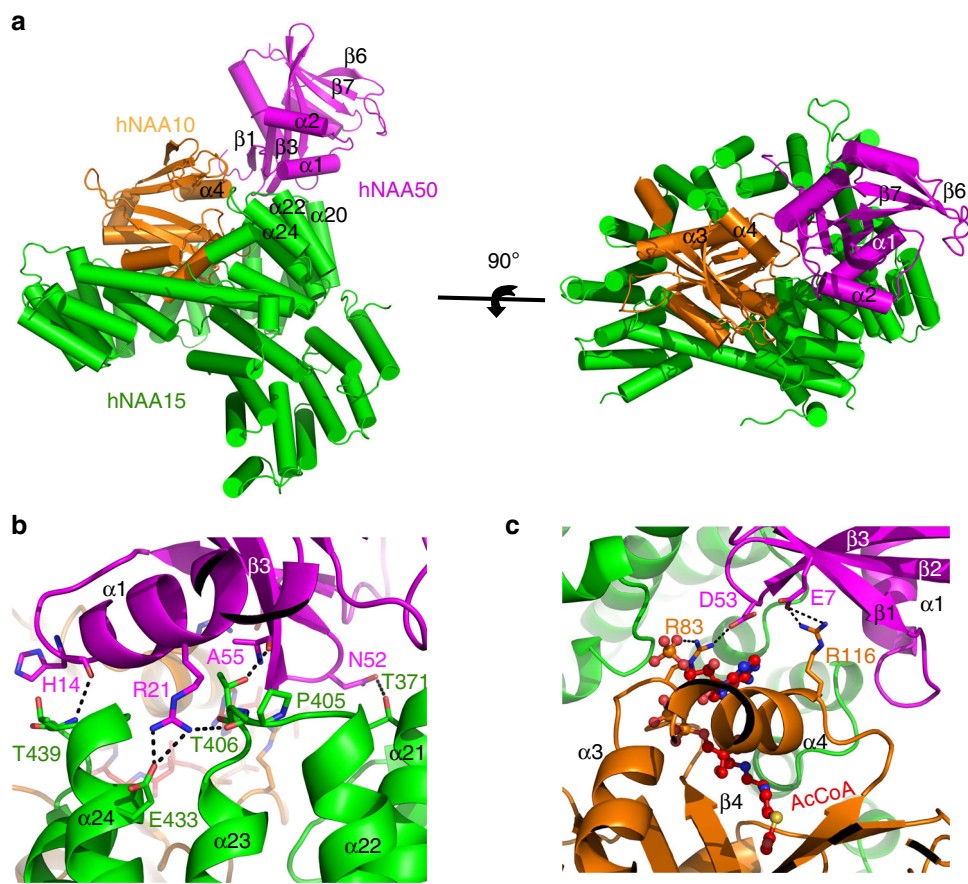

**Fig. 4 Cryo-EM structure of the hNatE complex. a** hNAA50(magenta), hNAA15(green), and hNAA10(orange) within the hNatE complex shown in cartoon. **b** Zoom-in view of the contacts between hNAA15 and hNAA50 with residues that participate in interaction labeled. **c** Zoom-in view of the contacts between hNAA10 and hNAA50 with residues that participate in interaction shown. Acetyl-CoA bound to hNAA10 is shown in ball and stick.

the observed weaker binding interaction between hNAA50 and hNatA in the presence of HYPK.

To understand the molecular basis for the biochemical findings that the binding of hNAA50 and HYPK to hNatA exhibit negative cooperativity, we superimposed the ternary hNatA/HYPK and hNatE complexes onto the hNatE/HYPK complex (Fig. 7). To align the hNatE and hNatE/HYPK complexes, we superimposed the hNAA50 position to delineate how the HYPK binding regions are affected by hNAA50 binding (Fig. 7a). The overall overlay suggests that both the hNAA10 and hNAA15 subunits of hNatA shift in the direction that is away from the hNAA50 binding region to make closer and more optimal contacts with HYPK (Fig. 7a). The hNAA15 α36–α40 helices, which bind to the C-terminal UBA domain (α3–α5) of HYPK in the tetrameric hNatE/HYPK complex, clearly shift away from the binding interface, when only hNAA50 is bound to hNatA (Fig. 7c). Notably, α40, α38, α37, and α36 display shift of ~4.5 Å, ~2.3 Å, ~1.6 Å, and ~1.8 Å, respectively. The important residues of hNAA15 (Lys696, Arg697, Lys687, and Lys685), which mediate HYPK interactions in this region suggested that most of them in the hNatE structure exhibit an ~2 Å shift away from HYPK (Fig. 7c). Meanwhile, hNAA15 N-terminal helices of α7, α8, α9, and α10, which are in close contact with HYPK-α2 also show ~3.0 Å, ~4.1 Å, ~1.9 Å, and ~3.5 Å shifts, respectively (Fig. 7d), while Tyr158 is about ~2 Å further away. Lastly, we observe that α2 and β3–β4 of hNAA50-bound hNAA10 move ~2.5 Å closer to contact the HYPK N-terminal α1 helix (Fig. 7e). Thus, it appears that hNAA50 binding to hNatA, destabilizes hNatA-HYPK interactions through moving hNatA away from its optimal binding position to HYPK.

Similarly, a superposition of hNatE/HYPK with the crystal structure of hNatA/HYPK by aligning to the HYPK position to delineate how the hNAA50 binding regions are affected by HYPK binding reveals that hNAA15 is shifted away from HYPK and closer to hNAA50 in the hNatE/HYPK complex (Fig. 7b). A further zoom-in view of the hNAA50 binding regions in hNAA15 demonstrates that residues Glu433, Thr406, Thr371, and Thr439 of hNAA15 are shifted by ~2 − 3 Å away from hNAA50 when HYPK is bound relative to their positions in the absence of hNAA50 (Fig. 7f). Regarding the hNAA50-hNAA10 binding interface, a 4.3 Å shift of hNAA10-Arg116 was observed, with a 1.7 Å for hNAA10-Arg83 (Fig. 7g). It is also noteworthy that significant conformational shift of the C-terminal hNAA15 helices α41–α45(~2.2 − 4.2 Å) was induced, in order to have hNAA50 bound in the tetrameric hNatE/HYPK complex (Fig. 7h). This suggests that HYPK bound hNatA is also not optimal for hNAA50 binding.

Taken together, these comparisons reveal that hNAA50 and HYPK destabilize the binding of the other protein to hNatA, despite their independent binding surfaces to hNatA, and explain the observed negative cooperative binding of hNAA50 and HYPK to hNatA.

**Molecular basis for decrease of hNatE activity by HYPK.** Our kinetic data demonstrated that the catalytic efficiency of hNatE is greater than hNAA50, but that this increase is partially nullified in the presence of HYPK (Table 1, Fig. 2b). To understand the molecular basis for this, we overlayed structures of hNatE and hNatE/HYPK with the common hNatA subunits

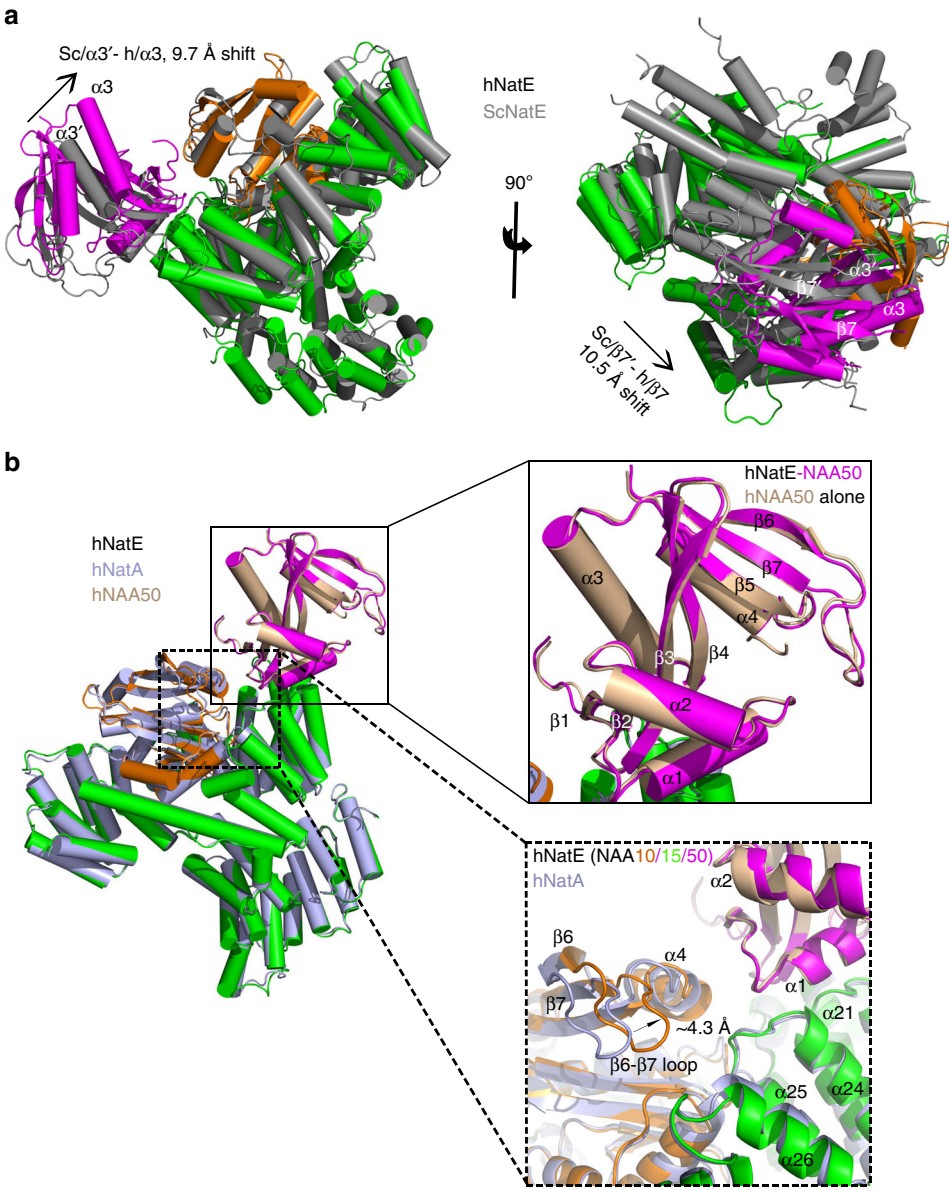

**Fig. 5 Subunit crosstalk within the hNatE complex. a** hNAA50(magenta), hNAA15(green), and hNAA10(orange) within the hNatE complex overlay with ScNatE (gray, PDB: 6O07). The α3 helix and β7 strand of Naa50 is shown to shift toward Naa10 in the human structure (**b**) hNatE aligned with hNatA (light blue, PDB:6C9M) and hNAA50 (wheat, PDB: 3TFY). The top zoom-in area shows the alignment of free hNAA50 and hNatE. The below zoom-in area shows the hNAA10 conformational change induced by hNAA50 binding.

aligned. This superposition reveals that the largest structural differences within the hNAA50 subunit map to the β6–β7 and α1–α2 loops, which shift such that the hNAA50 peptide substrate-binding groove is ~2.4 Å wider in the hNatE/HYPK complex relative to the hNatE complex (Fig. 8). In addition, the hNAA50 β3–β4 loop is shifted in the direction away from hNatA by ~3.5 Å in the presence of HYPK (Fig. 8). The previous studies demonstrated that Tyr73 and Met75 from hNAA50-β4 and Tyr138, Tyr139 and Leu142 from the hNAA50 β6-β7 loop are important for hNAA50 catalytic function and substrate binding[21]. To directly test this, we prepared the following mutants: hNAA50-Y73F, -M75A, -Y138A, -Y139A, and -I142A, and tested their effects on hNatE activity when HYPK was bound (Fig. 2d). Consistent with the previous studies, Y73F, Y138A, and Y139A decreased hNAA50 activity to undetectable levels[21]. For hNAA50-M75A and -I142A, we observed increased activity when bound to hNatA to form the

hNatE complex, as we observed for hNAA50-WT. As expected, the hNatE/HYPK complex displayed reduced activity compared to both hNAA50 alone and hNatE. However, we observed that hNatE-M75A/HYPK and hNatE-I142A/HYPK exhibited similar activity, compared to their corresponding hNAA50 mutants alone. These data are consistent with the conclusion that perturbation of the hNAA50 β3−β4 and β6−β7 loops within the hNatE/HYPK complex underlies the observed reduced catalytic efficiency of hNatE in the presence of HYPK.

## Discussion

While the interaction and catalytic crosstalk within NatE in the yeast system had previously been reported, the corresponding human system and the influence of HYPK binding on hNatE activity had not previously been characterized. Here, we biochemically and structurally characterize the mechanistic interplay of the hNAA10 and hNAA50 catalytic subunits within the hNatE

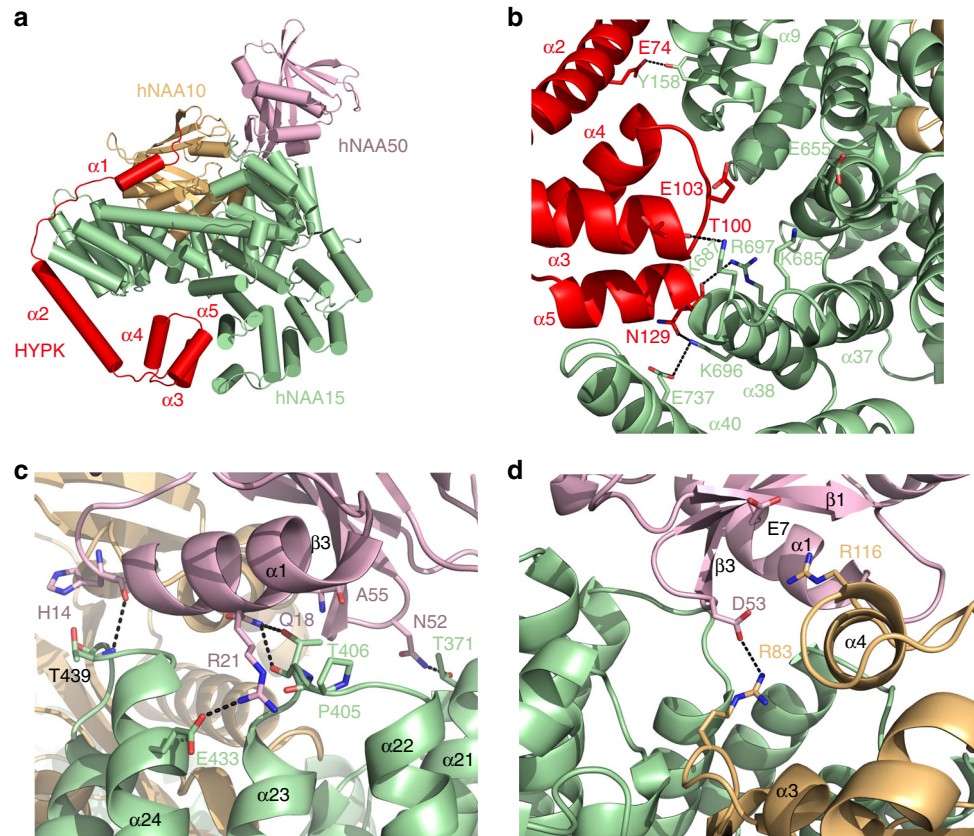

**Fig. 6 Overall structure of the hNatE/HYPK complex. a** hNAA50 (pink), hNAA15 (green), hNAA10 (orange), and HYPK (red) within the hNatE complex is shown in cartoon. **b** Zoom-in view of the contacts between HYPK and hNAA15 with residues that participate in interaction shown. **c** Zoom-in view of the contacts between hNAA15 and hNAA50 with residues that participate in interaction shown. **d** Zoom-in view of the contacts between hNAA10 and hNAA50 with residues that participate in interaction shown.

complex, and the role of HYPK in regulating hNatE activity. Similar to the crystal structure of ScNatE (PDB: 6O07), we find that hNAA50 binds hNatA mainly through hNAA15, using highly conserved residues, including a key threonine residue located at the center of the binding interface. Notably, the distance between hNAA50 and hNAA10 in the human complex is significantly shorter than in the yeast complex. We find that these two acidic hNAA50 residues, Glu7 and Asp53 contact two hNAA10 arginine residues, inducing a conformational change in hNAA10, which contributes to a decrease in hNatA enzymatic activity.

While both hNAA50 and hNatA are conserved from yeast to human, it had long been considered that HYPK is not present in yeast until the recently identified HYPK in thermophilic fungus *Chaetomium thermophilum*[44]. Nevertheless, it appears that evolutionarily NAA50 appeared as a NatA-binding partner earlier than HYPK did. In both budding and fission yeast, NAA50 is inactive but regulates NatA acetylation activity[46]. While in higher eukaryotes, NAA50 is enzymatically active, either with or without NatA[15,21]. Thus, within the higher eukaryotic tetrameric NatE/HYPK complex, it is likely that the regulation of NatA activity by NAA50 was replaced by HYPK.

Here, we have reconstituted the tetrameric hNatE/HYPK complex to demonstrate that HYPK and hNAA50 can bind to hNatA simultaneously. We also confirmed the physiological relevance of the tetrameric hNatE/HYPK complex through immunoprecipitation of the complex from HeLa cells. We find that while HYPK inhibits hNatA activity directly, it also indirectly, but less potently, inhibits hNatE. We also find that hNAA50 and HYPK exhibit negative cooperativity with respect to hNatA

binding in vitro and in vivo. The structure of the hNatE/HYPK complex suggests that this is due to the ability of the hNAA50 and HYPK protein to bind to the hNAA15 subunit of hNatA in a way that indirectly destabilizes binding of the other protein. This observation, together with our findings that HYPK has a dominant regulatory effect on hNatA activity relative to hNAA50 may explain why excess free hNAA50 is observed over hNatA in higher organism and HeLa cells[48].

The nanomolar dissociation constants of HYPK and hNAA50 for hNatA is consistent, with our immunoprecipitation of the tetrameric hNatE/HYPK complex from cells. The previous studies in *Drosophila* also reported that a dNAA50 knockout decreases in vivo dNatA catalytic activity in the presence of HYPK[45]. In terms of the biological function of this tetrameric complex, we propose two non-mutually exclusive regulatory mechanism for how hNAA50 protein levels could affect hNatA activity. First, since hNAA50 has a negative effect on binding between hNatA and HYPK, the absence of hNAA50, would promote HYPK binding to hNatA to inhibit hNatA activity. Second, since a recent yeast NatE-ribosome structure demonstrated that yeast NAA50 participates in ribosome association[49], it is likely that hNAA50 in higher eukaryotes also contributes to the association between tetrameric hNatE/HYPK and ribosome. The lower level of hNAA50 could therefore decrease the fraction of ribosome-associated hNatA to regulate its co-translational activity.

The overall structures of hNAA50 and ScNAA50 are well conserved with a mix of α-helices and β-strands. However, the overlay of hNatE and ScNatE suggested that NAA50 shifts closer to NAA10 in the human over the yeast complex. It is possible that in the

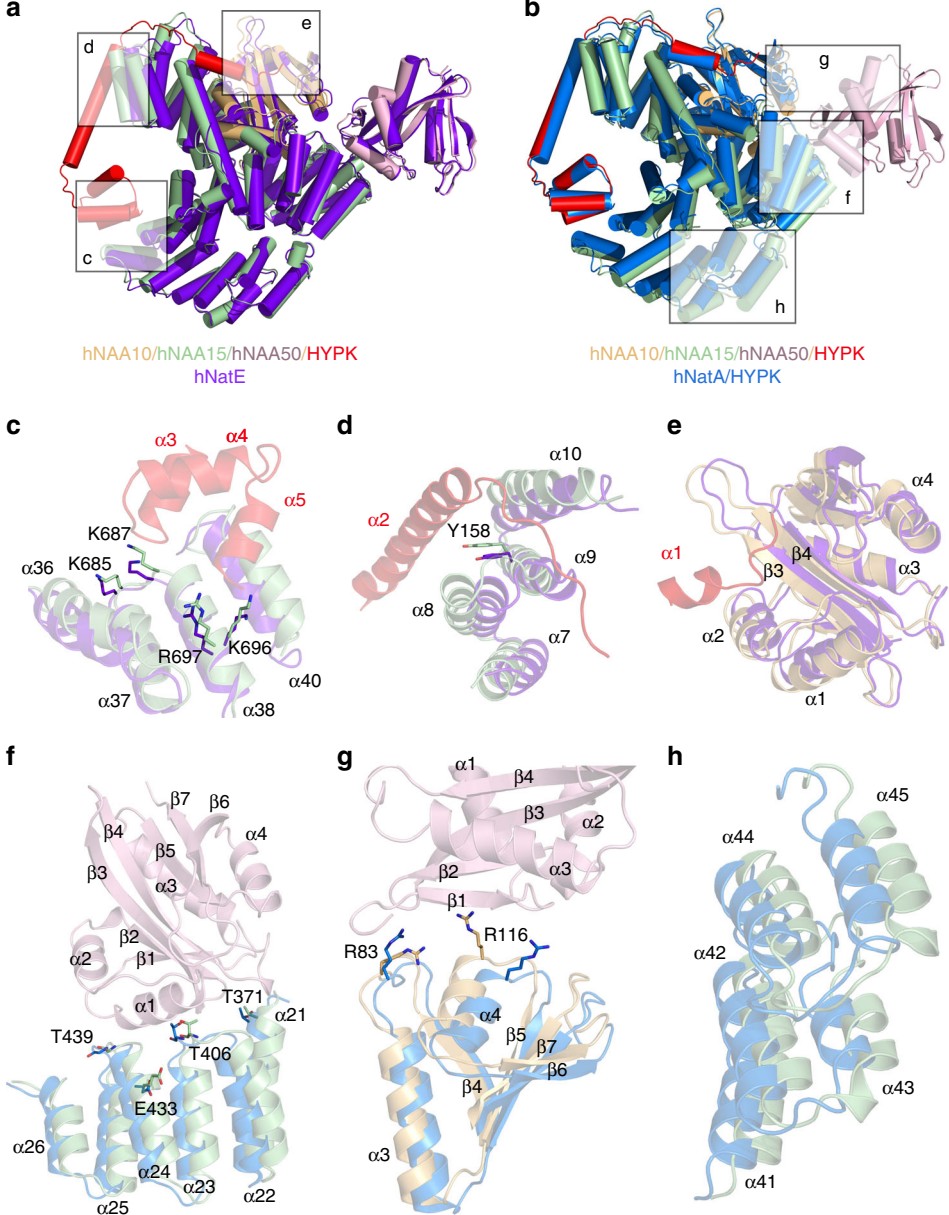

**Fig. 7 Molecular basis for hNAA50 and HYPK binding to hNatA. a** hNatE/HYPK overlayed onto hNatE (purple) with NAA50 aligned. **b** hNatE/HYPK overlayed onto hNatA/HYPK (PDB: 6C95, blue) with HYPK aligned. **c** Zoom-in view of HYPK C-terminal binding region as indicated in **a**. **d** Zoom-in view of HYPK α2 binding region as indicated in **a**. **e** Zoom-in view of HYPK N-terminal α1 domain binding region as indicated in **a**. **f** Zoom-in view shows the hNAA50 binding region on hNAA15 as indicated in **b**. **g** Zoom-in view shows the hNAA50 binding region on hNAA10 as indicated in **b**. **h** Zoom-in view shows the hNAA15 C-terminal helices conformal changes when HYPK bound as indicated in **b**.

human system NAA10 and NAA50 have more intimate contact and crosstalk in enzymatic activity. However, the previous data indicated that the degree of NAA50 and NAA10 activity crosstalk is similar in yeast and human[46]. In a recently reported yeast ScNatE-ribosome structure, the ScNAA10 and ScNAA50 catalytic sites are 50 Å and 85 Å away from the peptide exit site on the ribosome[49]. While unlike ScNAA50, hNAA50 is co-translationally enzymatically active[15]. Thus, we propose that this shift of NAA50 closer to NAA10 in the human over the yeast complex may have an impact on the relative distance between the nascent chain and active sites of hNAA10 and hNAA50.

With respect to hNatA binding to the ribosome, it is possible that HYPK may also contribute to ribosome binding by hNatA. This could then indirectly affect hNAA50 recruitment to the ribosome, which would be promoted by hNatA but inhibited by

HYPK. HYPK contains a UBA domain, which is also present at the C-terminus of the alpha subunit of the nascent polypeptide-associated complex. Further studies are required to test these hypotheses for hNatA recruitment to the ribosome.

The hNatE/HYPK complex co-translationally acetylates nascent peptides with N-termini of either methionine maintained or cleaved, which accounts for the largest number of substrates among all NATs. Given that this complex acetylates ~40–60% of the human proteome and has altered function in many human diseases, the studies presented here provides an important molecular scaffold for potential therapeutic development.

## Methods

**Plasmid construction**. A mammalian expression vector *pcDNA3.1/NAA15-V5* was modified by site-directed mutagenesis (Q5® Site-Directed Mutagenesis Kit, New

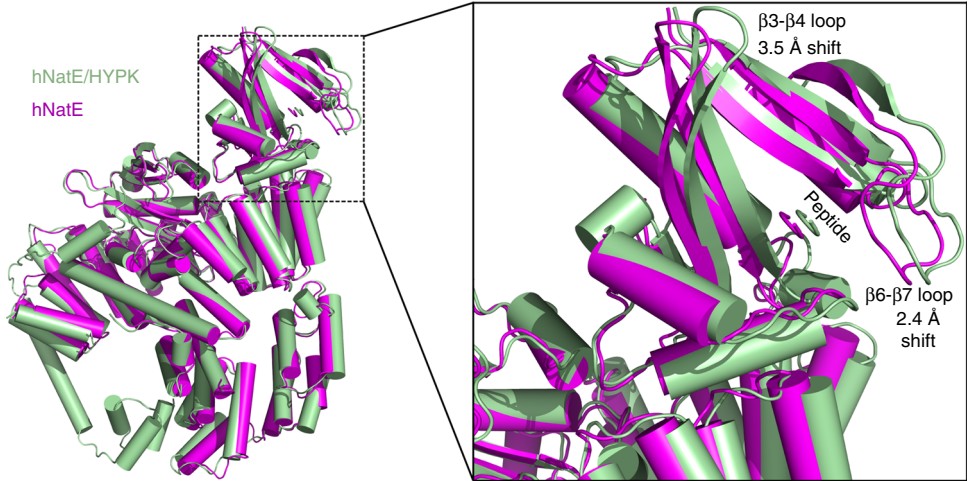

**Fig. 8 Molecular basis for the decrease of hNAA50 activity by HYPK.** hNatE/HYPK (green) overlayed onto hNatE (magenta) with hNatA aligned. Zoom-in view shows the local conformational change of hNAA50.

England Biolabs) to encode hNAA15 variants c.1216-1218ACA > TAC (p.T406Y) and c.2441 T > C (p.L814P). Primer sequences for p.T406Y are *AAGTACACCTTACTTA ATAGAACTCTTTCTCGTG* and *TCAATAGCAGTATTTATGTAC*. Primer sequences for p.L814P are *TGATGGTAGCCCAGGAGACTGTA* and *TACAAGGCTTCCAATA CC*. The mutated plasmid sequences were verified by sequencing.

**Proteomics sample preparation of NAA15-V5 immunoprecipitates.** For immunoprecipitation of NAA15-V5 variants, two 10 cm dishes of HeLa cells (ATCC CCL-2) were transfected with 10 μg of either *pcDNA3.1/NAA15-WT-V5*, *pcDNA3.1/ NAA15-T406Y-V5*, or *pcDNA3.1/NAA15-L814P-V5* using X-tremeGENE™ 9 DNA Transfection Reagent (Roche). The growth medium was replaced after 24 h. Forty eight hours post transfection, cells were harvested and lysed in 200 μl IPH lysis buffer (50 mM Tris-HCl pH 8.0, 150 mM NaCl, 5 mM EDTA, 0.5% NP-40, 1× complete EDTA-free protease inhibitor cocktail (Roche)) per cell dish for 15 min at 4 °C on a rotating wheel. Cell debris was pelleted by centrifugation (17,000 × g, 4 °C, 5 min) and the supernatant was mixed with 2 μg of V5-tag mouse monoclonal antibody (Invitrogen, R960-25) and incubated for 2 h at 4 °C on a rotating wheel. Thereafter, 20 μl of Dynabeads™ Protein G (Invitrogen) washed in IPH lysis buffer was added to the cell lysates and incubated overnight. The next day, beads were washed three times in IPH lysis buffer before the retrieved immunocomplexes were eluted in 60 μl FASP buffer (2% SDS, 100 mM Tris-HCl pH 7.6, and 0.1 M dithiothreitol (DTT)) and heated for 5 min at 95 °C. Filter-aided sample preparation (FASP) method was performed to process the eluates for liquid chromatography–MS/MS analysis[50]. The protein samples were mixed with 200 μl UA buffer (8 M urea, 100 Mm Tris-HCl pH 8.0), transferred to Microcon 30 kDa MWCO filters and centrifuged. All centrifugation steps were carried out at 23 °C and 14,000 × g for 15 min. The filters were washed three times with 200 μl UA by spinning. Proteins were then Cys-alkylated by incubation with 100 μl 50 mM iodoacetamide (IAA) in UA for 20 min, before IAA was removed by centrifugation. Filters were then washed three times with 100 μl UA. Finally, the UA buffer was exchanged by three washes with 100 μl 50 mM ammonium bicarbonate (ABC), before 500 ng trypsin (Sequencing Grade Modified Trypsin, Promega) was added to the filters and incubated overnight at 37 °C. Digested proteins were collected by centrifugation and the filter was washed once with 75 μl ABC. Peptides were acidified with 5% formic acid (FA) and desalted using Pierce™ C18-Tips (Thermo Scientific) according to manufacturer´s protocol. The final eluates were dried by speed vacuum and diluted to desired concentration with 5% FA.

**MS analysis of NAA15-V5 immunoprecipitates.** MS analysis was performed with an Ultimate 3000 RSLC system (Thermo Scientific) coupled to a Q-Exactive HF mass spectrometer (Thermo Scientific) equipped with EASY-spray nano-electrospray ion source (Thermo Scientific). About 1 μg tryptic peptides were loaded and desalted on a pre-column (Acclaim PepMap 100, 2 cm × 75 μm ID nanoViper column, packed with 3 μm C18 beads) with 0.1% trifluoroacetic acid (flow rate 5 μl/min, 5 min). Peptides were separated during a biphasic acetonitrile (ACN) gradient from two nanoflow UPLC pumps (flow rate of 200 nl/min) on an analytical column (PepMap RSLC, 50 cm × 75 μm i.d. EASY-spray column, packed with 2 μm C18 beads). Solvent A and B were 0.1% FA (vol/vol) in water and 100% ACN, respectively. The gradient composition was 5% B for 5 min, 5–8% B for 0.5 min, 8–24% B for 109.5 min, 24–35% B for 25 min, and 35–80% B for 15 min, 80% B over 15 min for isocratic elution and 5% B over 20 min for conditioning. The eluting peptides were ionized in the electrospray and analyzed by the Q-Exactive HF. The mass spectrometer was operated in data-dependent mode to automatically switch between full scan MS and MS/MS acquisition. MS spectra (m/z 375–1500) were obtained with a resolution of 120,000 at m/z 200, automatic gain control (AGC) target of 3 × 106 and maximum injection time

(IT) of 100 ms. The 12 most intense peptides above a threshold of 50,000 counts and charge states two to five were isolated (window of 1.6 m/z, AGC target of 1 × 105 and maximum IT of 110 ms) for fragmentation at a normalized collision energy of 28%. Fragments were detected in the orbitrap at a resolution of 15,000 at m/z 200, with first mass fixed at m/z 100. Precursor masses selected for MS/MS analysis were excluded by dynamic exclusion for 25 s with "exclude isotopes" enabled. Lock-mass internal calibration (m/z 445.12003) was used. The raw data acquired was processed with MaxQuant v. 1.6.2.6 and Andromeda search engine. The spectra were searched against a database of Swiss-Prot annotated human protein sequences (2,0431 sequences, retrieved 25.06.2019) and a reverse decoy database. Cystein carbamido-methylation was selected as a fixed modification and variable modifications included methionine oxidation and protein N-terminal acetylation. The false discovery rate was set to 1% for peptide and protein identification, minimum peptide length allowed was seven and match between runs (0.7 min match time window, 20 min alignment time window) was enabled. Label-free quantification[51] and IBAQ were selected. All other parameters were set to default values. The resulting proteingroups.txt file was analyzed using Perseus software v. 1.6.5.0. Proteins only identified by site, common contaminants and reverse hits were filtered away. The IBAQ intensities of hNAA10, hNAA50, and HYPK were normalized to the IBAQ intensity of NAA15 in each IP sample. Thereafter, the IBAQ intensity of hNAA15, hNAA10, hNAA50, and HYPK in each sample was normalized to the IBAQ intensity of the corresponding protein in the hNAA15-WT-V5 IP sample.

**[14C]-Ac-CoA-based acetylation and western blot analysis.** In order to test the NatA Nt-acetylation activity of hNAA15-T406Y and hNAA15-L815P variants, immunoprecipitation of hNAA15-V5 was performed as described above with minor adjustments. Five 10 cm dishes of HeLa cells (ATCC CCL-2) were transfected with 8 μg *pcDNA3.1/NAA15-WT-V5*, 8 μg *pcDNA3.1/NAA15-T406Y-V5*, or 10 μg *pcDNA3.1/NAA15-L814P-V5*. *NAA15-WT-V5* and *NAA15-T406Y-V5* were co-transfected with 2 μg of empty *pcDNA3.1/V5* vector to ensure equal conditions for the cells. As a negative control, five dishes of HeLa cells were transfected with 10 μg of *pcDNA3.1/LacZ-V5*. Immunoprecipitation from harvested and lysed cells was performed using 3 μg of V5-tag mouse monoclonal antibody (Invitrogen, R960-25) and 30 μl of Dynabeads™ Protein G (Invitrogen). The beads were washed three times in IPH lysis buffer and resuspended in 95 μl acetylation buffer (50 mM Tris-HCl pH 8.5, 1 mM EDTA, 10% Glycerol). [14 C]-Ac-CoA-based Nt-acetylation assays were performed as described[52]. In brief, three reaction mixture replicates were prepared containing 10 μl of immunoprecipitated enzyme, 200 μM synthetic 24-mer oligopeptide SESS24: NH2-SESSSKSRWGRPVGRRRRPVRVYP-COOH (BioGenes), 50 μM [14C]-Ac-CoA (Perkin-Elmer), and acetylation buffer to a final volume of 25 μl. Reaction mixtures without oligopeptide were used as negative controls. The reaction mixtures were incubated at 37 °C at 1400 rpm shaking for 30 min. The reaction was stopped by isolating beads on a magnet and transferring 23 μl of the supernatant onto P81 phosphocellulose filter discs (Millipore). The filter discs were washed three times in 10 mM HEPES buffer (pH 7.4) and air dried, before they were added to 5 ml Ultima Gold F scintillation mixture (Perkin-Elmer) and the incorporated [14C]-Ac was measured by a Perkin-Elmer TriCarb 2900TR Liquid Scintillation Analyzer. The immunoprecipitate input was determined by Western blot analysis. Proteins were separated by SDS–PAGE, transferred onto a nitrocellulose membrane (Amersham Protran 0.2 μM NC), and probed with V5-tag mouse monoclonal antibody (1:5000, Invitrogen, R960-25) and NAA10 rabbit monoclonal antibody (1:1000, Cell Signaling #13357). Protein bands were imaged by ChemiDoc™ XRS + system (Bio-Rad) and quantified using Imagelab™ Software (Bio-Rad). The measured disintegrations per minute (DPM)

signal for each reaction was normalized to the amount of NAA15-V5 in the respective IP sample.

**Protein expression and purification.** N-his-tagged hNatA and N-his-tagged hNatA/HYPK were expressed in sf9 cells (ThermoFisher, cat #12659017) and purified as described previously[16]. hNAA50[21] and MBP-HYPK[16] were expressed in BL21 (DE3) *Escherichia coli* cells (Thermo Scientific) and Rosetta (DE3)pLysS *E. coli* cells (Millipore Sigma), respectively, and purified as described previously[16]. For hNatA and hNatA/HYPK sf9 expression, high density ($2 \times 10^6$ cells ml$^{-1}$) suspension cultures of Sf9 cells were infected at a multiplicity of infection of 1 for 48 h in Fernbach Shake flasks at 27 °C. Cell pellets were resuspended in lysis buffer of 25 mM Tris, pH 8.0, 500 mM NaCl, 10 mM imidazole, 10 mM β-mercaptoethanol (β-ME), 10 mg ml$^{-1}$ PMSF (phenylmethanesulfonylfluoride), DNase, and complete, EDTA-free protease inhibitor tablet (Roche). After sonication, clarified lysate was passed on nickel resin (Thermo Scientific), washed with 10 column volumes (CV) of lysis buffer, and eluted with 25 mM Tris, pH 8.0, 200 mM NaCl, 200 mM imidazole, and 10 mM β-ME. The elutent was further purified on a HiTrap SP ion-exchange column with a salt gradient (200 mM to 1 M NaCl). Peak fractions were pooled and run on a Superdex 200 Increase 10/300 GL gel filtration column in sizing buffer containing 25 mM HEPES, pH 7.0, 200 mM NaCl, and 1 mM tris-2-carboxyethyl)phosphine (TCEP). Peak fractions were pooled and flash-frozen for storage in −80 °C until use.

For *E. coli* protein expression, transformed cells were cultured at 37 °C until the absorbance $A_{600}$ reached ~0.7, induced with 0.5 mM IPTG (isopropyl 1-thio-β-d-galactopyranoside), and grown overnight at 16 °C. *E.coli* cells overexpressing MBP-HYPK were lysed in 25 mM Tris, pH 8.0, 150 mM NaCl, 10 mM β-ME, and 10 mg ml$^{-1}$ PMSF. Clarified lysate was passed on to amylose agarose resin (New England Biolabs), washed with lysis buffer, and eluted in lysis buffer supplementaed with 20 mM maltose. Eluted protein was purified with a 5 ml HiTrap Q ion-exchange column (GE Healthcare) in the same buffer with a gradient (150 mM to 1 M NaCl). Peak fractions were pooled and loaded to a Superdex 75 Increase 10/300 GL gel filtration column (GE Healthcare) in buffer of 25 mM HEPES, pH 7.0, 200 mM NaCl, and 1 mM TCEP. Peak fractions were pooled and flash-frozen for storage in −80 °C until use. *E. coli* cells overexpressing GST-tagged NAA50 were lysed in 25 mM HEPES pH 7.5, 100 mM NaCl, and 10 mM β-ME by sonication. The supernatant was isolated and loaded to GST-binding resin (Clontech), washed with 10 CV lysis buffer. Tobacco etch virus protease was added to the resin for on-column cleavage overnight at room temperature. Untagged Naa50 was eluted from the column with lysis buffer and collected for overnight dialysis in buffer with 25 mM HEPES (pH 7.5), 50 mM NaCl, and 10 mM β-ME. Ion exchange was carried out with a 5-ml HiTrap SP ion exchange column (GE Healthcare) with a gradient of 50–750 mM NaCl. Peak fractions were further purified using a Superdex-75 gel filtration column (GE Healthcare) to homogeneity in buffer containing 25 mM HEPES, pH 7.5, 100 mM NaCl, and 10 mM DTT. Protein was flash-frozen for storage in −80 °C until use.

Protein harboring mutations was generated with the QuickChange protocol (Stratagene)[53] and obtained following the same expression and purification protocol as described for the WT protein. Primers for making mutations were as follows; hNAA50-E7A: *AAGGTAGCCGGATCGCGCTGGGAGATGTG* and *CAC ATCTCCCAGCGCGATCCGGCTACCTT*; hNAA50-D53A: *CTTGCCTATTTCAAT GCTATTGCTGTAGGTGC* and *GCACCTACAGCAATAGCATTGAAATAGGCAA G*; hNAA50-M75A: *GACTTTACATCGCGACACTAGGATG* and *CATCCTAGTGT CGCGATGTAAAGTC*; hNAA50-Y138A: *ACAAAGAAGAACGCCTATAAGAGGA TAG* and *CTATCCTCTTATAGGCGTTCTTCTTTGT*. The mutants hNAA5 0-Y73F, hNAA50-I142A, and hNAA50-Y139A were generated as previously described[21].

**Acetyltransferase activity assays.** Acetyltransferase assays were modified from previous studies[21,43] and carried out at room temperature in a reaction buffer containing 75 mM HEPES, pH 7.0, 120 mM NaCl, and 1 mM DTT. The SASE substrate peptide (NH$_2$-SASEAGVRWGRPVGRRRRP-COOH; GenScript) and the MLGP substrate peptide (NH$_2$-MLGPEGGRWGRPVGRRRRP-COOH; GenScript) were used to determine the enzymatic activity of hNatA and hNAA50, respectively. Each curve was repeated at least three times. To test the effect of hNAA50 and HYPK on hNatA activity, 100 nM of hNatA was mixed with 500 μM SASE peptide, 300 μM C$^{14}$ labeled acetyl-CoA (4 mCi mmol$^{-1}$; PerkinElmer Life Sciences), and varied concentrations of hNAA50 or HYPK modulator were added for 12-minute reactions. Signals were normalized against enzyme without modulator. For time course activity assays, 500 nM of hNAA50, hNatE, or hNatE/HYPK were mixed with 500 μM MLGP peptide, and 300 μM C$^{14}$-labeled acetyl-CoA (4 mCi mmol$^{-1}$; PerkinElmer Life Sciences) in a 20 μl reaction volume. Only 15 μl of each reaction mixture was quenched at specific times. For kinetic assays of hNAA50, hNatE and hNatE/HYPK against MLGP peptide, 300 nM enzyme was mixed with 300 μM C$^{14}$-labeled acetyl-CoA (4 mCi mmol$^{-1}$; PerkinElmer Life Sciences) and the concentration of the peptide substrate was varied for a 40-min reaction. Data were fit to a Michaelis–Menten equation in GraphPad Prism for determination of kinetic parameters.

**FP binding assays.** FP binding assays were performed essentially as described previously[46]. A total of 10 nM of fluorescein labeled hNAA50 (ref. [46]) was used in all reactions, and hNatA or hNatA/MBP-HYPK concentrations were varied to determine the $K_d$. A total of 5 mg ml$^{-1}$ bovine serum albumin and 0.2% v/v Tween were added into the reaction buffer (25 mM HEPES, pH 7.0, 200 mM NaCl, and 10 mM DTT) to prevent non-specific binding. FP readings were taken with a Perkin-Elmer EnVision and each curve was repeated in triplicate. GraphPad Prism (version 5.01) was used for all data fitting to determine $K_d$. A single-site specific binding model was used, with Eq. (1),

$$F_i = F_0 + F_1 \times \frac{K_d + [M] + [L] - \sqrt{(K_d + [M] + [L])^2 - 4 \times [M] \times [L]}}{2 \times [M]} \quad (1)$$

where $F_i$ is the fluorescence reading at ligand (either hNatA or hNatA/HYPK) concentration $[L]_i$; $K_d$ is the equilibrium dissociation binding constant; $[M]$ is the concentration of hNaa50; $F_0$ is the fluorescence reading extrapolated to no ligand; and $F_1$ is the maximum fluorescence increase at saturating ligand concentration. Fit parameters for hNatA/HYPK(MBP tagged) binding were $F_0 = 161.3 \pm 3.3$, $F_1 = 222.0 \pm 5.2$, $K_d = 127 \pm 13$ nM, and $R^2 = 0.98$. For hNatA, $F_0 = 205.8 \pm 4.4$, $F_1 = 137.2 \pm 5.4$, $K_d = 46 \pm 8.8$ nM, and $R^2 = 0.95$.

**ITC measurements.** ITC measurements were carried out using a MicroCal iTC200 at 20 °C. Samples were dialyzed into buffer containing 25 mM HEPES pH 7.0, 200 mM NaCl, and 1 mM DTT. Protein samples (hNatA and hNatE) with concentrations of 15 μM in the cell and 150 μM of MBP-HYPK in the syringe were used in the experiments. The raw data were analyzed with the MicroCal ITC analysis software.

**Cryo-EM sample preparation and data collection.** To prepare hNatE complex, purified hNatA and three molar excess of hNAA50 were mixed and loaded onto a Superdex 200 10/30 GL column (GE Healthcare). Peak fractions with all subunits present, as confirmed with SDS–PAGE analysis, were concentrated to 1 mg ml$^{-1}$. Fresh sample incubated with three molar excess of both hNatA and hNAA50 bi-substrate analogs for ~30 min on ice was applied to Quantinfoil R1.2/1.3 holey carbon support grids. To prepare hNatE/HYPK complex, purified hNatA/HYPK and three molar excess of hNAA50 were mixed and loaded onto an S200 gel filtration column (GE Healthcare). Peak fractions with all subunits present, as confirmed with SDS–PAGE were concentrated to 1 mg ml$^{-1}$. Fresh sample with three molar excess of acetyl-CoA and hNAA50 bi-substrate analogs for ~30 min in ice was applied to Quantinfoil R1.2/1.3 holey carbon support grids.

Both sample grids were blotted for 10–12 s (blot force = 2) under 100% humidity at 16 °C before the sample was plunged into liquid ethane, using a FEI Vitrobot Mark IV. An FEI TF20 was used for screening the grids and data collection was performed either with a Talos Arctica microscope or Titan Krios equipped with a K2 Summit direct detector (Gatan).

**Cryo-EM data processing.** Original image stacks were summed and corrected for drift and beam-induced motion at the micrograph level using MotionCor2 (ref. [54]). Defocus estimation and the resolution range of each micrograph were performed with Gctf[55]. For hNatA/hNAA50, ~3000 particles were manually picked to generate several rough two-dimensional (2D) class averages. Representative 2D classes were used to autopick ~1,229,331 particles from 4025 micrographs using Relion[56]. All particles were extracted and binned to accelerate the 2D and three-dimensional (3D) classifications. After bad particles were further removed by 2D and 3D classification, 353,541 particles were used for auto refinement, particle polishing, and per particle contrast transfer function (CTF) refinement. The final map of the hNatE complex was refined to an overall resolution of 3.02 Å, with local resolution estimated by Resmap[57]. For the hNatE/HYPK complex, image processing workflow was similar as described above. A total of 477,608 particles were picked from 1004 micrographs and 168,536 particles were used for refinement. The final resolution of the hNatE/HYPK complex was 4.03 Å.

**Cryo-EM model building and refinement.** For hNatE model building, the crystal structures of hNatA (PDB: 6C9M) and hNAA50(PDB: 3TFY) were fit into the 3.02 Å EM map as rigid bodies, followed by manual adjustment in coot[58] and real-space refinement in PHENIX[59]. For the hNatE/HYPK model building, the crystal structure of hNatA/HYPK (PDB: 6C95) and hNAA50 (PDB: 3TFY) were used. All representations of cryo-EM density and structural models were performed with Chimera[60] and PyMol (The PyMOL Molecular Graphics System, Version 1.2r3pre, Schrödinger, LLC).

**Reporting summary.** Further information on research design is available in the Nature Research Reporting Summary linked to this article.

## Data availability
The cryo-EM map for the hNatE and hNatE/HYPK and the respective atomic coordinates have been deposited in the EMDataBnak and ProteinDataBank, with accession codes of EMD-20442 and 6PPL for hNatE, respectively, and with EMD-20501 and 6PW9 for hNatE/HYPK, respectively. The mass spectrometry datasets have been deposited to the ProteomeXchange/PRIDE member repository under the PDX accession

code PXD017031. The source data for Table 1, Figs. 1b, 2a–d, 3a, b, and Supplementary Fig. 2 are provided in the Source Data file.

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

## Acknowledgements

This work was supported by NIH grant R35 GM118090 awarded to R.M. T.A was supported by the Research Council of Norway (Project 249843), the Norwegian Health Authorities of Western Norway (Project 912176), the Norwegian Cancer Society, and the European Research Council (ERC) under the European Union Horizon 2020 Research and Innovation Program under grant agreement 772039. Parts of the work was carried out at the Proteomics Unit at University of Bergen (PROBE). We acknowledge the support of the Perelman School of Medicine, University of Pennsylvania DNA Sequencing Core Facility and E. Dean from the High-Throughput Screening Core Facility for providing the Sf9 cells containing human NatE for this study. We thank Zuo Biao and Sudheer Molugu from Electron Microscopy Resource Lab of University of Pennsylvania for the help of initial cryo-grids screening; Chen Xu, KangKang Song, and Kyounghwan Lee from the University of Massachusetts Medical School Cryo-EM Core Facility for technical assistance on data collection; and Dan Ricketts, Adam Olia, Austin Vogt, and Leah Gottlieb for helpful discussions.

## Author contributions

Conceptualization, S.D. and R.M.; methodology, S.D., N.M., X.W., T.A. and R.M.; investigation, S.D., N.M. and X.W.; formal analysis, S.D., N.M. and X.W.; writing—original draft, S.D.; visualization, S.D.; writing—review and editing, S.D., N.M., X.W., T.A. and R.M.; funding acquisition; T.A. and R.M.; resources, T.A. and R.M.; supervision, T.A. and R.M.

## Competing interest

The authors declare no competing interests.
