## [Peer Review File · Nature Communications]

Reviewers' comments:

Reviewer #1 (Remarks to the Author):

Interaction and crosstalk within the human NatE and the influence of the Huntingtin yeast two-hybrid protein K (HypK) binding on hNatE have not been previously characterized. The work of Deng et al., allows a significant step forward, by providing mechanistic insights into the reciprocal regulation of the heterodimeric human NatE complex by the different catalytic components (hNaa50 and hNaa10) and the role of HypK in the regulation of the hNatE activity. Using biochemical studies and cryo-EM structures, the authors demonstrate that hNatE and HypK form a stable tetrameric complex. Deng et al. however show that HypK and Naa50 negatively affect their mutual binding with respect to hNatA. The authors provided also several lines of evidence suggesting that hNaa50 and HypK inhibit hNatA activity, with HypK inhibition that overwhelms the inhibitory effect of hNaa50.

The authors also show that the activity of NatE is higher than that of hNaa50, but this improvement is abolished by HypK. Finally, the authors provide cryo-EM structures of hNatE in complex with AcetylCoA and different substrate analogues. Although hNaa50 predominantly interacts with the auxiliary subunit Naa15, few interactions are observed also between hNaa50 and hNaa10 and suggested to contribute to the catalytic cross-talk between the two enzymes. Finally, the cryo-EM structure of tetrameric hNatE/hypK suggests the molecular basis of how Naa50 and HypK negatively cooperate in binding hNatA. Most of the experiments appear carefully performed. Nonetheless, few issues need to be addressed and clarify to strengthen the paper:

Major issues

- 1) One of my major concerns is related to the physiological meaning of the stable tetrameric complex involving both hNatE and HypK. As clearly demonstrated by the authors, HypK and hNaa50 display negative cooperative binding with respect to hNatA and influencing hNatA activity as well as hNatE activity. The authors should raise this important question at least in the conclusion.
- 2) Unlike HypK whose NatA inhibition is complete at concentration above 100 nM, no complete inhibition by hNaa50 is observed even at high concentration of the modulator. The author should give a possible explanation of hNaa50 behaviour at least in the discussion session.
- 3) In the result session concerning the cryo-EM structure and particularly the overlay of the Sc and Hs NatE (page 8), the authors observe that Naa50 shifts closer to Naa10 in the human over the yeast complex and involves alpha3 and beta7 of hNaa50. Thus, the authors claim that this more intimate contact in the human system could manifest in greater differences when these proteins are bound to the ribosome. However, I lack the logic of this statement and I do not understand what the authors exactly want to deliver as message. This whole paragraph should be part of the discussion (no results are presented in favour of a possible hypothesis) and more clearly re-written.

4) From superimposition of the hNatE structure with the hNatA structure the authors reveal a displacement of Naa10 beta6-beta7 loop towards hNaa50 (page 8), proposing that this shift plays a more significant role in the inhibitory effect on hNatA activity.

Since no data are presented in support of this hypothesis, I suggest to move this paragraph in the discussion or to envisage ad hoc mutants to test the hypothesis.

5) Unfortunately, the authors' hNatE structure and its superimposition with hNaa50 structure doesn't reveal any significant structural changes and provide no clue to explain at molecular level the increase catalytic efficiency of hNatE compared to hNaa50.

6) The authors proposed that since i) a shift of hNaa50-beta6-beta7 and hNaa50 beta3-beta4 loop is observed in the presence of HypK and ii) Tyr73, Met75, Tyr138, Tyr139 and Leu142 belonged to hNaa50 beta4 and beta6-beta7 loop were previously shown important for hNaa50 catalytic activity, that the perturbation of these interactions in the presence of HypK are responsible of the reduced catalytic efficiency. The authors need to prove this statement using mutants of these residues and demonstrate that HypK inhibition is abolished.

Minor issues:

1) Abstract: the authors claim that "NatE contains the Naa50 catalytic subunit with substrate specificity for most of the acetylated methionine-containing proteins". This statement is incorrect due to NatB and NatC. The author should be more precise in their formulation, particularly in the abstract.

2) In the introduction the authors claims the existence of several Nats (A-H) but they do not make any statements about NatG

3) Unlike HypK whose inhibition on NatA is complete at concentration above 100 nM, no complete inhibition is observed by hNaa50 even at high concentration of the modulator. The author should give a possible explanation at least in the discussion session.

4) Not clear to me how the experiment of Figure 2a related to hNatA inhibition by both HypK and hNaa50 has been performed. Which concentration of both proteins has been used?

5) As pointed out by the authors the NatA significantly enhances the catalytic efficiency of hNatE, largely through a decrease in the K_m . Indeed, the k_{cat} (rate of turnover) of both enzymes is almost the same. Then, I would not claim that the two enzymes have different activity. Please, being more precise throughout the text.

6) They are few typos errors in the text. Please, pay attention to them.

Reviewer #2 (Remarks to the Author):

“Molecular basis for N-terminal acetylation by human NatE and its modulation by HYPK”

Deng, Wei and Mamorstein provide structural insights into the human NatE (hNaa15/hNaa10/hNaa50) and a human NatE/HYPK (hNaa15/hNaa10/hNaa50/hHYPK) complex by cryo-EM. Through comparison of the presented cryo-EM structures with previously published crystal structures of NatE complex from *Saccharomyces cerevisiae* and NatA (Naa15/Naa50)/HYPK complexes from humans and *Chaetomium thermophilum* they demonstrate the evolutionary conservation of the overall architecture of these complexes. By fluorescence polymerisation measurements, they show that HYPK and hNaa50 might display negative cooperative binding with respect to hNatA. By isothermal titration calorimetry they determine Kds for the interaction of a MBP-HYPK fusion protein with either the hNatA (Naa15/Naa10) or hNatE (Naa15/Naa10/Naa50) complex. In acetylation assays, they show that HYPK inhibits both catalytic subunits hNAA10 and hNaa50. Using the structural information, the authors identify crucial amino acid residues for subunit interaction in hNaa50 and show their influence on acetylation activities of Naa10.

While the study provides the first human NatE (hNaa15/hNaa10/hNaa50) and hNatE/hHYPK structures it provides little new insight into the structural arrangement of these complexes or acetylation of nascent polypeptide chains by NatA or NatE, which essentially happens cotranslationally. The overall arrangement of the 4 protein components in the in vitro-reconstituted tetrameric complex is highly similar to previously published structures of trimeric complexes of the different components from different species (Weyer et al., 2017 Nature Communications, 15726; Gottlieb & Mamorstein, 2018, Structure 26, 925-935; Deng et al, 2019, Structure 27, 1-14). Whereas the determined partial structural rearrangements within the subunits might cause the observed influences on enzymatic activity and subunit interaction, the manuscript lacks convincing evidence for the in vivo existence of the tetrameric complex. In addition, the provided biochemical and biophysical experiments are not entirely convincing (see below).

In summary, the manuscript in the current form does not provide the level of novelty and common interest, which would justify publication in Nature Communications and I therefore suggest publication in a more specialized journal. In this case, the following points should be addressed:

“hNatA, hNatB, hNatC are three major NATs, each consisting of a catalytic subunit and one (hNatA and hNatB) or two (hNatC) auxiliary subunits¹¹.”

The provided reference is from 2003 and does not reflect the current state of knowledge.

Page 2 line 21

“hNatE contains hNatA and an additional catalytic subunit, hNaa50^{12,13}.”

Both references describe Yeast N-acetyl transferases and cannot be cited for the composition of the human complex.

For *S. cerevisiae* it was already shown in 2003 that NatA is composed of three subunits (Nat1/Ard1 and Nat5) or according to the new unified nomenclature (Naa10/ Naa15 and Naa50). Gautschi et al. (2003) *Mol Cell Biol* 23, 7403-7414.

Page 3 line 15

“Previous studies have demonstrated that HYPK harbors intrinsic NatA activity.....”. This implies that HYPK can acetylate proteins? Please explain.

In a previous publication the authors used the term NatE for the independently active Naa50 protein alone (Liszczyk & Mamorstein, *PNAS*, 2013). It is somewhat confusing for a general audience to follow the subunit composition of the different complexes throughout this manuscript.

Fig. 1 b

The binding curves should reach a plateau, they cannot decline again ? Please explain.

Page 4 line 20

“This data demonstrates that HYPK can negatively affect stability of the hNatE complex.” At the same time the header of the previous paragraph states: “hNatE and HYPK forms a stable tetrameric complex”.

Fig. 1c

Please add information about fit data e.g. ΔH , ΔG , $-T\Delta S$, stoichiometry etc.

Page 4 line 25

“The reason for the apparent two-binding transition when HYPK is titrated into hNatA is unknown, but we speculate that it is likely caused by the bipartite binding of the HYPK C-terminal and N terminal regions to hNatA with different affinities.”

The ITC measurements are performed with a MBP-HYPK fusion protein, which implies that the N-terminus of HYPK is not freely available for its known interaction with the Naa10 active site. This might be the reason for the apparent two-binding transition. A control of NatA plus MBP alone should be provided. Such a two-binding transition was not observed by Weyer et al. Nature Communications 8, 15726 for the *C. thermophilum* HYPK which also harbors two binding sites.

Fig 1 d

The curve does not reach a plateau. The right side of the upper panel appears as if there is buffer mixing, indicative of a scenario where the two components in the cell and in the syringe are not in the same buffer. The molar ratio is not 1 as in Fig. 1c. Please discuss.

According to the Methods section in this experiment 15 μ M of hNatA or hNatE in the cell and 150 mM MBP-HYPK in the syringe were used which is a 10,000 fold excess. That would equal roughly 8.7g/ml MBP-HYPK! At this concentration difference, the reaction should be finished after the first injection?

Fig.2a

Please label in the figure that MBP-HYPK and not HYPK was used.

hNaa50 inhibits NatA by \sim 50% already at Naa50 concentrations way below 1 μ M (compare to text page 5 line 16)

Fig. 2c

Data for hNatE and hNaa50 appear suspiciously identical to data already published in Fig. 4 b in Deng et al. 2019, Structure 27, 1-14.

Page 6 line 23

“After appropriate adjustment of the model,.....” Please explain.

In general, the structural data presented appear very sound but please provide a supplementary Fig. showing a fit of the model in the electron density.

Supplementary Fig. 1: Please provide information about the resolution at which the presented volumes were filtered.

Supplementary Fig. 2: One possible reason for the lower local resolution of the HYPK in the Cryo-EM structure could be an underrepresentation of HYPK in the complex. Compare Fig. 1d.

Page 14 line5

“In most yeast, Naa50 is inactive but regulates NatA acetylation activity.” Please provide a reference for this statement.

Page 15 line 8

“This is consistent with the presence of a nascent chain ribosome association (NAC) domain within HYPK14.”

HYPK does not contain a NAC domain. It does contain a UBA domain, which is also present at the C-terminus of the alpha subunit of the Nascent Polypeptide-Associated Complex (NAC). The UBA domain is not involved in NAC's association with the ribosome.

Response to Reviewers:

Reviewer #1

Interaction and crosstalk within the human NatE and the influence of the Huntingtin yeast two-hybrid protein K (HypK) binding on hNatE have not been previously characterized. The work of Deng et al., allows a significant step forward, by providing mechanistic insights into the reciprocal regulation of the heterodimeric human NatE complex by the different catalytic components (hNaa50 and hNaa10) and the role of HypK in the regulation of the hNatE activity. Using biochemical studies and cryo-EM structures, the authors demonstrate that hNatE and HypK form a stable tetrameric complex. Deng et al. however show that HypK and Naa50 negatively affect their mutual binding with respect to hNatA. The authors provided also several lines of evidence suggesting that hNaa50 and HypK inhibit hNatA activity, with HypK inhibition that overwhelms the inhibitory effect of hNaa50. The authors also show that the activity of NatE is higher than that of hNaa50, but this improvement is abolished by HypK. Finally, the authors provide cryo-EM structures of hNatE in complex with AcetylCoA and different substrate analogues. Although hNaa50 predominantly interacts with the auxiliary subunit Naa15, few interactions are observed also between hNaa50 and hNaa10 and suggested to contribute to the catalytic cross-talk between the two enzymes. Finally, the cryo-EM structure of tetrameric hNatE/hypK suggests the molecular basis of how Naa50 and HypK negatively cooperate in binding hNatA. Most of the experiments appear carefully performed. Nonetheless, few issues need to be addressed and clarify to strengthen the paper:

Major issues

1) One of my major concerns is related to the physiological meaning of the stable tetrameric complex involving both hNatE and HypK. As clearly demonstrated by the authors, HypK and hNaa50 display negative cooperative binding with respect to hNatA and influencing hNatA activity as well as hNatE activity. The authors should raise this important question at least in the conclusion.

To confirm the physiological relevance of the tetrameric hNatE/HYPK complex, we immunoprecipitated c-terminal V5 tagged hNAA15 from HeLa cells to evaluate the associated proteins. Mass spectrometry analysis of this sample revealed that endogenous hNAA10, hNAA50 and HYPK were all co-immunoprecipitated with NAA15-WT-V5, in agreement with a tetrameric hNatE/HYPK complex. We also carried out immunoprecipitation with hNAA15-T406Y-V5 and hNAA15-L814P-V5 mutants that are compromised in NAA50 and HYPK binding, respectively, and showed that these mutants destabilized binding to the corresponding protein. Finally, we evaluated the hNatA activity of the immunoprecipitated complexes, with their activities corresponding to our *in vitro* results. This new data is now described in a new results section entitled "Binding properties of hNatA to NAA50 and HYPK is recapitulated in human cells" and illustrated in new Fig. 3, Supplementary Table 1 and Supplementary Fig. 2.

2) Unlike HypK whose NatA inhibition is complete at concentration above 100 nM, no complete inhibition by hNaa50 is observed even at high concentration of the modulator. The author should give a possible explanation of hNaa50 behaviour at least in the discussion session.

We have now revised the text to more clearly state that unlike HYPK, NAA50 mediates only partial inhibition of NatA up to a concentration of 1 μ M, although the reason for this remains unclear. We would prefer not to speculate on this point.

3) In the result session concerning the cryo-EM structure and particularly the overlay of the Sc and Hs NatE (page 8), the authors observe that Naa50 shifts closer to Naa10 in the human over the yeast complex and involves alpha3 and beta7 of hNaa50. Thus, the authors claim that this more intimate contact in the human system could manifest in greater differences when these proteins are bound to the ribosome. However, I lack the logic of this statement and I do not understand what the authors exactly want to deliver as message. This whole paragraph should be part of the discussion (no results are presented in favour of a possible hypothesis) and more clearly re-written.

We agree with the reviewer and have moved this more speculative part of the manuscript into the discussion section of the manuscript.

4) From superimposition of the hNatE structure with the hNatA structure the authors reveal a displacement of Naa10 beta6-beta7 loop towards hNaa50 (page 8), proposing that this shift plays a more significant role in the inhibitory effect on hNatA activity. Since no data are presented in support of this hypothesis, I suggest to move this paragraph in the discussion or to envisage ad hoc mutants to test the hypothesis.

We agree with the reviewer and have now softened the text to propose that “this shift in position also contributes to the inhibitory effect that hNaa50 binding has on hNatA activity.”

5) Unfortunately, the authors' hNatE structure and its superimposition with hNaa50 structure doesn't reveal any significant structural changes and provide no clue to explain at molecular level the increase catalytic efficiency of hNatE compared to hNaa50.

We have indeed not seen any structural change or rearrangement of hNAA50 within hNatE, thus we hypothesize that the increase activity of hNAA50 might be due to a reduced entropic cost for substrate binding to the hNaaAA50 subunit due to hNatA tethering, and we now direct readers to references that support this proposal.

6) The authors proposed that since i) a shift of hNaa50-beta6-beta7 and hNaa50 beta3-beta4 loop is observed in the presence of HypK and ii) Tyr73, Met75, Tyr138, Tyr139 and Leu142 belonged to hNaa50 beta4 and beta6-beta7 loop were previously shown

important for hNaa50 catalytic activity, that the perturbation of these interactions in the presence of HypK are responsible of the reduced catalytic efficiency. The authors need to prove this statement using mutants of these residues and demonstrate that HypK inhibition is abolished.

In the revised manuscript, we prepare these mutants and demonstrate that while the hNaa50-Y73F, Y138A and Y139A decreased hNAA50 activity to undetectable levels, the hNatA-M75A and -I142A mutants retained increased activity of hNatA when bound to hNAA50 to form the hNatE complex, as we observed for hNatA-WT. Significantly, in the presence of HYPK, the hNatA-M75A and -I142A mutants exhibited similar activity, compared to their corresponding hNAA50 mutants alone. This data is consistent with the conclusion that the perturbation of the hNAA50 β 3– β 4 and β 6– β 7 loops within the hNatE/HYPK complex underlies the observed reduced catalytic efficiency of hNatE in the presence of HYPK. This new data is now described in the results section "Molecular basis for the decrease of hNatE activity by HYPK" and shown in new figure panel 2d.

Minor issues:

1) Abstract: the authors claim that "NatE contains the Naa50 catalytic subunit with substrate specificity for most of the acetylated methionine-containing proteins". This statement is incorrect due to NatB and NatC. The author should be more precise in their formulation, particularly in the abstract.

We have now removed this statement and revised the text to simply indicate that NatE has specificity for N-terminal methionine acetylation.

2) In the introduction the authors claims the existence of several Nats (A-H) but they do not make any statements about NatG.

We have now added information on NatG.

3) Unlike HypK whose inhibition on NatA is complete at concentration above 100 nM, no complete inhibition is observed by hNaa50 even at high concentration of the modulator. The author should give a possible explanation at least in the discussion session.

As we indicated in response of Major Issue 2, we have now revised the text to more clearly state that unlike HYPK, NAA50 mediates only partial inhibition of NatA up to a concentration of 1 μ M, although the reason for this remains unclear. We would prefer not to speculate on this point.

4) Not clear to me how the experiment of Figure 2a related to hNatA inhibition by both HypK and hNaa50 has been performed. Which concentration of both proteins has been used?

We have added text to the figure legend to indicate that “Either MBP-HYPK, hNAA50, or both are titrated into hNatA (100nM) to test their modulatory effect on hNatA activity against an SASE peptide substrate.

5) As pointed out by the authors the NatA significantly enhances the catalytic efficiency of hNatE, largely through a decrease in the K_m . Indeed, the k_{cat} (rate of turnover) of both enzymes is almost the same. Then, I would not claim that the two enzymes have different activity. Please, being more precise throughout the text.

We have revised the text to more accurately state that two enzymes have different catalytic efficiencies (K_{cat}/K_m).

6) They are few typos errors in the text. Please, pay attention to them.

We have carefully reviewed the revised manuscript to correct typos.

Reviewer #2

“Molecular basis for N-terminal acetylation by human NatE and its modulation by HYPK” Deng, Wei and Mamorstein provide structural insights into the human NatE (hNaa15/hNaa10/hNaa50) and a human NatE/HYPK (hNaa15/hNaa10/hNaa50/hHYPK) complex by cryo-EM. Through comparison of the presented cryo-EM structures with previously published crystal structures of NatE complex from *Saccharomyces cerevisiae* and NatA (Naa15/Naa50)/HYPK complexes from humans and *Chaetomium thermophilum* they demonstrate the evolutionary conservation of the overall architecture of these complexes. By fluorescence polymerisation measurements, they show that HYPK and hNaa50 might display negative cooperative binding with respect to hNatA. By isothermal titration calorimetry they determine K_d s for the interaction of a MBP-HYPK fusion protein with either the hNatA (Naa15/Naa10) or hNatE (Naa15/Naa10/Naa50) complex. In acetylation assays, they show that HYPK inhibits both catalytic subunits hNAA10 and hNaa50. Using the structural information, the authors identify crucial amino acid residues for subunit interaction in hNaa50 and show their influence on acetylation activities of Naa10.

While the study provides the first human NatE (hNaa15/hNaa10/hNaa50) and hNatE/hHYPK structures it provides little new insight into the structural arrangement of these complexes or acetylation of nascent polypeptide chains by NatA or NatE, which essentially happens cotranslationally. The overall arrangement of the 4 protein components in the in vitro-reconstituted tetrameric complex is highly similar to previously published structures of trimeric complexes of the different components from different species (Weyer et al., 2017 Nature Communications, 15726; Gottlieb & Mamorstein, 2018, Structure 26, 925-935; Deng et al, 2019, Structure 27, 1-14). Whereas the determined partial structural rearrangements within the subunits might cause the

observed influences on enzymatic activity and subunit interaction, the manuscript lacks convincing evidence for the *in vivo* existence of the tetrameric complex. In addition, the provided biochemical and biophysical experiments are not entirely convincing (see below).

In summary, the manuscript in the current form does not provide the level of novelty and common interest, which would justify publication in Nature Communications and I therefore suggest publication in a more specialized journal. In this case, the following points should be addressed:

We agree with the reviewer that the most novel aspect of the manuscript is the tetrameric NatE/HYPK complex. We also believe that the findings that HYPK indirectly inhibits NAA50 activity through the tetrameric complex and the negative cooperative binding among the subunits are novel. To further support the *in vitro* findings, we now present new supporting cell-based studies. Specifically, we immunoprecipitated c-terminal V5 tagged hNAA15 from HeLa cells to evaluate the associated proteins. Mass spectrometry analysis of this sample revealed that endogenous hNAA10, hNAA50 and HYPK were all co-immunoprecipitated with NAA15-WT-V5, in agreement with a tetrameric hNatE/HYPK complex. We also carried out immunoprecipitation with hNAA15-T406Y-V5 and hNAA15-L814P-V5 mutants that are compromised in NAA50 and HYPK binding, respectively, and showed that these mutants destabilized binding to the corresponding protein. Finally, we evaluated the hNatA activity of the immunoprecipitated complexes, with their activities corresponding to our *in vitro* results. This new data is now described in a new results section entitled “Binding properties of hNatA to NAA50 and HYPK is recapitulated in human cells” and illustrated in new Fig. 3, Supplementary Table 1 and Supplementary Fig. 2.

1) Page 2 line 19

“hNatA, hNatB, hNatC are three major NATs, each consisting of a catalytic subunit and one (hNatA and hNatB) or two (hNatC) auxiliary subunits¹¹.”

The provided reference is from 2003 and does not reflect the current state of knowledge.

We now provide a more recent reference:

Aksnes, H., Ree, R. & Arnesen, T. Co-translational, Post-translational, and Non-catalytic Roles of N-Terminal Acetyltransferases. *Mol. Cell* **73**, 1097-1114 (2019).

2) Page 2 line 21

“hNatE contains hNatA and an additional catalytic subunit, hNaa50^{12,13}.”

Both references describe Yeast N-acetyl transferases and cannot be cited for the composition of the human complex.

For *S. cerevisiae* it was already shown in 2003 that NatA is composed of three subunits (Nat1/Ard1 and Nat5) or according to the new unified nomenclature (Naa10/ Naa15 and

Naa50). Gautschi et al. (2003) Mol Cell Biol 23, 7403-7414.

We have now removed the human (h) designation and thus refer to the yeast complex, where NatE was first identified.

3) Page 3 line 15

“Previous studies have demonstrated that HYPK harbors intrinsic NatA activity.....”. This implies that HYPK can acetylate proteins? Please explain.

We have revised the text to now read “Previous studies have demonstrated that HYPK may be important for cellular NatA activity”

4) In a previous publication the authors used the term NatE for the independently active Naa50 protein alone (Liszczak & Mamorstein, PNAS, 2013). It is somewhat confusing for a general audience to follow the subunit composition of the different complexes throughout this manuscript.

In the introduction we added text to clarify this confusion. We state “ The trimeric hNAA10/hNAA15/hNAA50 complex is referred to as hNatE, and....”

5) Fig. 1 b

The binding curves should reach a plateau, they cannot decline again? Please explain.

We have refitted the data in Fig 1b and generated new binding curves, and the Kd numbers are updated.

6) Page 4 line 20

“This data demonstrates that HYPK can negatively affect stability of the hNatE complex.” At the same time the header of the previous paragraph states: “hNatE and HYPK forms a stable tetrameric complex”.

We have changed the header to now read “hNatE and HYPK forms a tetrameric complex.”

7) Fig. 1c

Please add information about fit data e.g. ΔH , ΔG , $-T\Delta S$, stoichiometry etc.

This information has now been added to revised Figure panel 1d.

8) Page 4 line 25

“The reason for the apparent two-binding transition when HYPK is titrated into hNatA is unknown, but we speculate that it is likely caused by the bipartite binding of the HYPK C-terminal and N terminal regions to hNatA with different affinities.”

The ITC measurements are performed with a MBP-HYPK fusion protein, which implies

that the N-terminus of HYPK is not freely available for its known interaction with the Naa10 active site. This might be the reason for the apparent two-binding transition. A control of NatA plus MBP alone should be provided. Such a two-binding transition was not observed by Weyer et al. Nature Communications 8, 15726 for the *C. thermophilum* HYPK which also harbors two binding sites.

A control run of MBP titrating into hNatA was performed and the curve is now shown in Supplementary Fig 1. We have now deleted our statement about the bipartite binding hypothesis. It is noteworthy that in the curve below (A screen shot of the ITC curve of by Weyer et al. Nature Communications 8, 15726 for the *C. thermophilum* HYPK which also harbors two binding sites) appears to also contain a small degree of bipartite binding.

9) Fig 1 d

The curve does not reach a plateau. The right side of the upper panel appears as if there is buffer mixing, indicative of a scenario where the two components in the cell and in the syringe are not in the same buffer. The molar ratio is not 1 as in Fig. 1c. Please discuss. According to the Methods section in this experiment $15\mu\text{M}$ of hNatA or hNatE in the cell and 150 mM MBP-HYPK in the syringe were used which is a 10.000 fold excess. That would equal roughly 8.7g/ml MBP-HYPK! At this concentration difference, the reaction should be finished after the first injection?

The titrant (MBP-HYPK) concentration in the syringe was actually $150\mu\text{M}$, and not 150 mM as indicated. This was an error as “m” was not converted to symbol font the accurately read “ μ ”. We were also perplexed about why the stoichiometry of MBP-HYPK titrated into hNatE is not equal to 1, but we suspect that the addition of excess NAA50 to NatA may render some NatA molecules incapable of binding HYPK. This could be

caused by a secondary NAA50 binding sites on NatA, that blocks HYPK binding. Correlating with this possibility, the cell-based mass spectrometry analysis of the hNatE-L814P mutant also shows sub-stoichiometric binding of HYPK together with super-stoichiometric binding of NAA50 (Fig. 3a). Since this is somewhat speculative, we have not included this discussion in the manuscript.

10) Fig.2a

Please label in the figure that MBP-HYPK and not HYPK was used.

This is now labeled in the revised figure.

11) hNaa50 inhibits NatA by ~ 50% already at Naa50 concentrations way below 1 μ M (compare to text page 5 line 16)

We have now revised the text to more clearly state that unlike HYPK, NAA50 mediates only partial inhibition of NatA up to a concentration of 1 μ M, although the reason for this remains unclear. We would prefer not to speculate on this point.

12) Fig. 2c

Data for hNatE and hNaa50 appear suspiciously identical to data already published in Fig. 4 b in Deng et al. 2019, Structure 27, 1-14.

This data is similar but not identical. In any regard, we have removed that data, but the calculated kinetic parameters are listed in Table 1.

13) Page 6 line 23

“After appropriate adjustment of the model,.....” Please explain.

This statement is now clarified in the methods section to indicate that, the model was manually adjustment in Coot and then using Phenix real space refined.

14) In general, the structural data presented appear very sound but please provide a supplementary Fig. showing a fit of the model in the electron density.

This is now shown in supplementary Figs. 6c and 6d.

15) Supplementary Fig. 1: Please provide information about the resolution at which the presented volumes were filtered.

The overall resolution limits of 3.02 and 4.03 Å for hNatE and hNatE/HYPK, respectively are now indicated in Supplementary Figure 5.

16) Supplementary Fig. 2: One possible reason for the lower local resolution of the HYPK in the Cryo-EM structure could be an underrepresentation of HYPK in the complex.

Compare Fig. 1d.

We feel that this is unlikely since particles that do not contain HYPK would fall into a different 3D class average.

17) Page 14 line5

“In most yeast, Naa50 is inactive but regulates NatA acetylation activity.” Please provide a reference for this statement.

We have modified the sentence to read that in both budding and fission yeast , NAA50 is inactive and have provided a reference where we have confirmed this:

Deng, S. *et al.* Structure and Mechanism of Acetylation by the N-Terminal Dual Enzyme NatA/Naa50 Complex. *Structure*,1057-1070(2019).

18) Page 15 line 8

“This is consistent with the presence of a nascent chain ribosome association (NAC) domain within HYPK14.”

HYPK does not contain a NAC domain. It does contain a UBA domain, which is also present at the C-terminus of the alpha subunit of the Nascent Polypeptide-Associated Complex (NAC). The UBA domain is not involved in NAC’s association with the ribosome.

We have revised the text as suggested.

Reviewers' comments:

Reviewer #1 (Remarks to the Author):

In the revised manuscript, Deng & al. have carefully addressed all concerns raised. The authors have provided a clearer text where the speculative parts have been rewritten more clearly and most displaced in the discussion section. Furthermore, the authors provided new ad hoc experiments including:

- 1) data showing that hNAA10, hNAA50 and HYPK were all co-immunoprecipitated with NAA15- WT-V5, confirming the existence of the tetrameric hNatE/HYPK complex in vivo, while the use of specific NAA15 mutants has shown a destabilization binding for the corresponding protein ;
- 2) molecular basis for the decrease of hNatE activity by HYPK, characterizing ad hoc mutants ;
- 3) a control run of MBP titrating into hNatA, showing a bipartite binding. While the reason of this bipartite curve is unclear the authors pointed out that this type of curve had already been observed also previously by others.
- 4) New refitted data.

In my opinion, the revised manuscript is now ready for publication in Nature Comms.

Reviewer #2 (Remarks to the Author):

Since highly similar structures from other species have been available for some time, my main concern with the manuscript by Deng et al. was the lack of novelty in the light of missing in vivo data that this complex is physiologically relevant, and some issues with the biophysical and biochemical characterization of the complexes.

In the revised manuscript, the authors made an effort to provide in vivo data based on immunoprecipitations, showing that the complex indeed exist. They also tested some mutations confirming their conclusions derived from the structural analysis, and adjusted their biochemical data.

To that end the novelty aspect is in principle sufficiently addressed.

However, I still have concerns regarding re-interpretation of some biochemical data:

For Figure 1b I pointed out in (5) of the evaluation that the binding data don't reach a plateau, which may be indicative of problems with the assay. The authors state now that they 'refitted the data', apparently adding data points to the same data (although claiming in the legend $n=3$ repetitions for the error bars in the old and also the revised version), now fitting a curve with a plateau, but also providing surprisingly different K_d values: i.e. 160 nM versus 120 nM for hNatA/HYPK.

More importantly, in point (12) of my evaluation I mentioned for Figure 2c that identical data for the Michaelis-Menten kinetic curves for hNaa50 and hNatE have been published before in Deng et al., 2019, Structure, with an absolutely identical distribution of data points (see attachment). I am very surprised by the author's statement that these data are 'similar but not identical', which is not accurate since they clearly 're-used' the data for hNaa50 and hNatE, and just added the new data for HypK. To me this completely undermines the author's credibility.

In light of the above mentioned inconsistencies I remain highly irritated but would like to leave it to the editor to assess whether or not the data are solid and trustworthy enough for publication in Nature Communications.

Responses to Reviewer 2

1. For Figure 1b I pointed out in (5) of the evaluation that the binding data don't reach a plateau, which may be indicative of problems with the assay. The authors state now that they 'refitted the data', apparently adding data points to the same data (although claiming in the legend $n=3$ repetitions for the error bars in the old and also the revised version), now fitting a curve with a plateau, but also providing surprisingly different K_d values: i.e. 160 nM versus 120 nM for hNatA/HYPK.

Both the old and the new curves are fit and generated in "GraphPad Prism 5" based on the same data set with $n = 3$. In the initial draft manuscript, we used a "One site-Total" model, in which the equation fits total binding only, assuming it is a sum of specific binding plus a linear nonspecific component. The K_d reported in the initial draft were calculated using this model. In addition, the data in the initial draft was plotted as "mean and error", and thus did not show each replicate. Please see the initial fitting in the figure below. Indeed, as the reviewer points out, the curves do not reach a plateau.

Figure 1b from initial draft manuscript.

In the revised manuscript, we fit the same data to a "single site binding" model. These new fitting curves generate slightly different K_d values as updated in the revised manuscript. We believe that the new K_d values are more accurate since the new fittings do indeed reach a plateau (see below). In the revised manuscript, we also replotted the data as "each replicate" as opposed to "mean and error" as suggested in the *Nature Communications* guidelines. This replotted data is also shown below using "each replicate" as in the revised manuscript or using "mean and error" for comparison (below). We apologize if plotting the data as "each replicate" may have given the reviewer the false impression that we added additional data.

Figure 1b from revised manuscript.

Figure 1b from revised manuscript but plotted as “mean and error”

In the revised manuscript, we have now added a paragraph to the methods section explicitly stating that a single site binding model was used for the fit and have also defined the equation employed for this model.

2. More importantly, in point (12) of my evaluation I mentioned for Figure 2c that identical data for the Michaelis-Menten kinetic curves for hNaa50 and hNatE have been published before in Deng et al., 2019, *Structure*, with an absolutely identical distribution of data points (see attachment). I am very surprised by the author’s statement that these data are ‘similar but not identical’, which is not accurate since they clearly ‘re-used’ the data for hNaa50 and hNatE, and just added the new data for HypK. To me this completely undermines the author’s credibility.

The assay in question was carried out independently twice (Experiments #1 and #2), each in triplicate, prior to publication of the Deng et al. 2019 *Structure* manuscript. In the Deng et al. 2019. *Structure* manuscript, Experiment 1 was used for Figure 4b of that manuscript, but the data for the hNatE/HYPK complex was not included since HYPK was not part of the story that was reported in *Structure*. For the initial draft manuscript, we did indeed inadvertently use the same data from Experiment 1 (as reported in Figure 4b of the Deng et al. 2019 *Structure* manuscript) but with data for the hNatE/HYPK complex included, and we thank the reviewer for catching this error in our initial manuscript. However, for the revised manuscript, we used the data from Experiment 2. Not surprisingly, this data is highly similar but not identical. The raw data for Experiment #1 (Columns D, E and F) and Experiment #2

(Columns A, B and C), the corresponding kinetic parameters calculated (using GraphPad Prism 5) and the superimposed data, to demonstrate that they are not identical, are shown below. In the revised manuscript, we now only report the calculated kinetic parameters from Experiment #2 in Table 1.

X	A			B			C			D			E			F			G
	hsNaa50(this manuscript)	hsNatA/E (this manuscript)	hsNatA/E/HYPK(this manuscript)	hNaa50 (structure paper)	hNatA/E (structure paper)	hNatA/E/HYPK (structure paper)	hNaa50 (this manuscript)	hNatA/E (this manuscript)	hNatA/E/HYPK (this manuscript)	hNaa50 (structure paper)	hNatA/E (structure paper)	hNatA/E/HYPK (structure paper)	hNaa50 (this manuscript)	hNatA/E (this manuscript)	hNatA/E/HYPK (this manuscript)				
1000.000000	3542	4341	4280	5866	6002	5889	1391	1534	1696	3844	3674	3634	4957	5887	6017	1420	1436	1514	
500.000000	2940	3017	2899	5574	5848	5818	1460	1601	1450	2399	2327	2316	5097	5534	5646	1208	1319	1366	
250.000000	1719	1729	1665	5109	5097	4784	1248	1351	1317	1452	1466	1294	4780	4774	5194	1151	1234	1151	
125.000000	943	923	770	3780	3943	3926	917	1135	1154	799	799	790	3745	3823	3859	1076	1069	1044	
62.500000	456	527	502	2238	2823	2639	704	775	780	434	441	399	2769	2622	2840	693	762	845	
31.250000	293	299	263	1544	1609	1370	455	451	445	255	241	238	1420	1458	1461	460	509	511	
15.625000	156	170	166	887	876	965	336	306	317	124	146	122	840	951	920	303	257	328	
7.812500	108	85	91	410	452	425	212	203	214	65	89	79	433	469	505	147	159	170	
3.906250	49	67	69	269	255	262	141	153	162	42	63	56	279	248	265	116	139	135	
1.953125	38	52	63	154	144	172	98	120	101	43	31	33	158	133	145	80	88	100	

Nonlin fit	A	B	C	D	E	F
	hsNaa50(this manuscript)	hsNatA/E (this manuscript)	hsNatA/E/HYPK(this manuscript)	hNaa50 (structure paper)	hNatA/E (structure paper)	hNatA/E/HYPK
1 Michaelis-Menten	Y	Y	Y	Y	Y	Y
2 Best-fit values						
3 Vmax	7516	6711	1683	8148	6303	1503
4 Km	831.5	96.18	73.11	1202	85.06	60.37
5 Std. Error						
6 Vmax	481.6	95.28	39.53	290.3	130.2	25.05
7 Km	93.73	4.533	6.062	67.39	5.996	3.696
8 95% Confidence Intervals						
9 Vmax	6529 to 8502	6516 to 6906	1602 to 1763	7554 to 8743	6037 to 6570	1452 to 1554
10 Km	639.6 to 1023	86.90 to 105.5	60.70 to 85.53	1064 to 1340	72.78 to 97.34	52.80 to 67.94
11 Goodness of Fit						
12 Degrees of Freedom	28	28	28	28	28	28
13 R2	0.9900	0.9951	0.9826	0.9982	0.9889	0.9903
14 Absolute Sum of Squa	533000	709142	155532	72606	1.481e+006	72713
15 Syx	138.0	159.1	74.53	50.92	230.0	50.96
16 Constraints						
17 Km	Km > 0.0	Km > 0.0	Km > 0.0	Km > 0.0	Km > 0.0	Km > 0.0
18 Number of points						
19 Analyzed	30	30	30	30	30	30

In the revised manuscript, we now indicate on the legend of Table 1 that calculation of kinetic parameters for hNAA50 and hNatE is based on an independent replicate of experiments previously reported and we reference Deng et al. 2019, *Structure* (reference 46).